# Predicting Future Utility: Global Combinatorial Optimization for Task-Agnostic KV Cache Eviction

**Ziyao Tang** [* 1 2]   **Pengkun Jiao** [* 1]   **Xinhang Chen** [3]   **Wei Liu** [3]   **Shiyong Li** [3]   **Jingjing Chen** [1]

## Abstract

Given the quadratic complexity of attention, KV cache eviction is vital to accelerate model inference. Current KV cache eviction methods typically rely on instantaneous heuristic metrics, implicitly assuming that score magnitudes are consistent proxies for importance across all heads. However, this overlooks the heterogeneity in predictive fidelity across attention heads. While certain heads prioritize the *instantaneous contribution* of tokens, others are dedicated to capturing *long-horizon utility*. In this paper, we propose that optimal budget allocation should be governed by the marginal utility in preserving long-term semantic information. Building on this insight, we propose **LU-KV**, a novel framework that formulates head-level budget allocation as a global combinatorial optimization problem to maximize the long-horizon marginal contribution of reserved tokens. To solve this non-convex problem, we employ a convex-hull relaxation and a marginal-utility-based greedy solver, achieving near-optimal solutions. Furthermore, we implement a data-driven offline profiling protocol to facilitate the practical deployment of LU-KV. Evaluations on Long-Bench and RULER benchmarks demonstrate that LU-KV reduces KV cache size by 80% with minimal performance degradation, while also decreasing inference latency and GPU memory footprint.
 Project Code

## 1. Introduction

The recent push towards long-context processing in Large Language Models (LLMs) is fundamentally bottlenecked by the Key-Value (KV) cache. As sequence lengths scale to millions of tokens, the linear explosion in memory consumption severely throttles inference throughput and hinders scalable deployment. Consequently, KV cache eviction (Zhang et al., 2023; Feng et al., 2026b; Kim et al., 2026) has emerged as an essential technique. It conventionally operates under a two-stage paradigm: *intra-head scoring* to filter critical tokens, and *cross-head budget allocation* to distribute the memory budget across attention heads.

While significant progress has been made in designing sophisticated scoring metrics (Zhang et al., 2023; Li et al., 2024), budget allocation strategies remain a critical yet underdeveloped frontier. Existing methods largely rely on instantaneous heuristic scoring, assuming that current attention magnitudes serve as reliable proxies for future importance. However, we identify a fundamental flaw in this magnitude-based paradigm: it ignores the inherent heterogeneity in predictive fidelity across different attention heads. Specifically, high-magnitude scores in certain heads often align poorly with **Oracle Importance**—the true long-term contribution to the KV cache—capturing transient noise rather than enduring semantic anchors. Blindly biasing budgets toward regions with high-magnitude heuristic scores, without accounting for their long-term utility, inevitably leads to suboptimal cache retention.

We posit that optimal budget allocation should be governed not by absolute scores, but by the marginal utility of a metric in preserving future information. In this view, memory allocation is treated as a strategic investment: if a metric exhibits poor alignment with the Oracle Importance in a specific head, increasing its budget yields rapidly diminishing returns. Conversely, in heads where the metric is precise, a unit investment in budget effectively preserves the model's long-horizon generative quality. Therefore, the crux of an efficient allocation strategy lies in quantifying and optimizing the long-term cost-effectiveness of each head as guided by a given metric.

Based on this insight, we propose **Long-horizon Utility KV (LU-KV)**, a novel framework for head-wise KV cache budget allocation. We introduce a data-driven offline calibration mechanism to profile the marginal contribution curves of individual attention heads. To construct these profiles, we formulate the global budget distribution as a combinatorial optimization problem aimed at maximizing the expected long-horizon utility retention across all heads. To solve this efficiently, we employ a convex-hull relaxation and a greedy solver, ensuring near-optimal budget allocation with minimal computational overhead.

---

[*]Equal contribution [1]Fudan University [2]Work done during an internship at Baidu [3]Baige AI Team, Baidu inc. Correspondence to: Jingjing Chen <chenjingjing@fudan.edu.cn>.

*Proceedings of the 43rd International Conference on Machine Learning*, Seoul, South Korea. PMLR 306, 2026.

Extensive experiments on the LongBench and RULER benchmarks demonstrate the effectiveness of LU-KV. Our method achieves an 80% reduction in KV cache size with minimal performance degradation.

Our contributions are summarized as follows:

- We identify a critical gap between heuristic importance metrics and long-horizon marginal utility in head-wise KV eviction, as these metrics fail to reflect the true long-horizon marginal contribution of tokens in certain heads.
- We formulate budget allocation as a long-term utility maximization problem and introduce an efficient solver using convex-hull relaxation and a marginal-utility-based greedy strategy. Furthermore, we propose an offline profiling protocol for practical deployment.
- We conduct extensive evaluations across diverse long-context benchmarks to validate the effectiveness and robustness of our proposed methods.

## 2. Related Work

Existing research on KV cache eviction can be broadly categorized into two synergistic streams: *intra-head eviction policies*, which identify informative tokens within individual heads, and *cross-head budget allocation*, which manages resource distribution across the entire model.

**Intra-head Eviction Policies**  Intra-head strategies focus on designing high-fidelity proxy metrics to distinguish critical tokens from noise. Early heuristics, such as StreamingLLM (Xiao et al., 2024), identified the "attention sink" phenomenon, showing that retaining initial tokens is crucial for maintaining model stability. Subsequently, methods like H2O (Zhang et al., 2023) and SnapKV (Li et al., 2024) utilized accumulated attention scores or observation windows to dynamically cluster and retain salient tokens. Beyond raw attention weights, recent literature has explored geometric and perturbation-based indicators to mitigate inherent biases. For instance, KeyDiff (Park et al., 2025) leverages the geometric features of Key vectors, while CriticalKV (Feng et al., 2025) explicitly measures potential output perturbations by considering Value magnitudes and projection weights. DefensiveKV (Feng et al., 2026a) adopts a worst-case view for robust eviction by aggregating scores within individual heads. KVZip (Kim et al., 2026) investigates query-agnostic KV cache compression via context reconstruction, using a reconstruction objective to guide cache eviction without relying on the current query. Despite these advances, existing methods primarily optimize token selection within a fixed per-head budget, overlooking how that budget should be allocated across heads.

**Cross-Head Budget Allocation**  Recognizing the heterogeneity of information density across layers, recent studies have shifted toward non-uniform allocation strategies. Static and rule-based methods often rely on structural priors; for example, PyramidKV (Cai et al., 2024) employs a fixed pyramidal shape based on the "information funneling" hypothesis, while HeadKV (Fu et al., 2025) and CAKE (Qin et al., 2025) incorporate task-specific priors or spatial dispersion to formulate cascading rules. In contrast, dynamic allocation strategies like Ada-KV (Feng et al., 2026b) attempt to distribute resources based on real-time statistics, such as attention entropy. However, these approaches inherently assume that proxy scores are well-calibrated and comparable across different heads—an assumption that often fails in practice due to varying score scales and metric inaccuracies.

Unlike previous methods that rely on instantaneous heuristic scoring and consequently overlook the long-horizon importance of tokens, we propose an allocation framework governed by long-horizon utility. Rather than directly comparing uncalibrated proxy scores, our approach profiles the budget-utility relationship to explicitly quantify the marginal gain of retaining tokens within each attention head. Furthermore, our framework is metric-agnostic: given any chosen proxy metric (*e.g.*, SnapKV), we can derive the optimal budget configuration to maximize long-horizon information retention.

## 3. Preliminaries

We consider decoder-only LLMs with $L$ layers and $H$ attention heads per layer. Inference proceeds in two phases: parallel prefill and autoregressive decoding.

### 3.1. Attention Mechanism and KV Cache

At decoding step $k$, for a fixed layer $\ell$ and head $h$, the model generates a query $\mathbf{q}_{\ell,h,k} \in \mathbb{R}^{d_h}$ to attend to historical keys $\mathbf{k}_{\ell,h,j}$ and values $\mathbf{v}_{\ell,h,j}$ ($j \leq k$). The attention weights $A$ and the head output are computed as:

$$A_{\ell,h,k,j} = \text{Softmax}\left( \frac{\mathbf{q}_{\ell,h,k}^{\top}\mathbf{k}_{\ell,h,j}}{\sqrt{d_h}} \right), \qquad (1)$$

$$\mathbf{o}_{\ell,k} = \sum_{h=1}^{H}\left( \sum_{j=1}^{T} A_{\ell,h,k,j}\mathbf{v}_{\ell,h,j} \right)\mathbf{W}_O^{(\ell,h)}. \qquad (2)$$

where $\mathbf{W}_O^{(\ell,h)} \in \mathbb{R}^{d_h \times d_{\text{model}}}$ is the head-specific output projection. To avoid redundant computation, previously computed $(\mathbf{k}, \mathbf{v})$ pairs are stored in a **KV Cache**. As the sequence length grows, the linear increase in cache size poses a significant memory bottleneck.

## 3.2. KV Cache Eviction

For readability, the main symbols used below are summarized before the formal setup.

| Symbol | Description |
|---|---|
| $L, H$ | Number of layers and number of attention heads per layer, respectively. |
| $(\ell, h)$ | The $h$-th attention head in the $\ell$-th layer. |
| $T$ | The total number of input tokens during the prefill phase. |
| $\pi$ | The metric used to evaluate token importance. |
| $\pi^*$ | The oracle metric of token importance. |
| $\sigma$ | Target compression ratio, i.e., the fraction of token positions evicted. |
| $B_{\text{total}}$ | Global all-head memory budget; for length $T$, $B_{\text{total}} = (1 - \sigma)L \times H \times T$. |
| $b_{\ell,h}$ | Memory budget allocated to a specific attention head $(\ell, h)$. |
| $r_{\ell,h}$ | Local head-level compression ratio, with $b_{\ell,h} = \lfloor (1 - r_{\ell,h})T \rfloor$. |
| $\mathcal{M}_{\ell,h}^{\pi}$ | Token indices at head $(\ell, h)$, sorted by $\pi$ (descending). |
| $\mathcal{M}_{\ell,h}^{\pi}(k)$ | Retained subset containing the top-$k$ elements of $\mathcal{M}_{\ell,h}^{\pi}$. |

To maintain a manageable memory footprint, *KV Cache Eviction* strategies (Feng et al., 2026b; Park et al., 2025) impose a fixed budget $B$ per head. These methods typically use attention scores as an importance proxy, retaining a subset of indices $\mathcal{I}_{\ell,h} \subset \{1, \ldots, T\}$ such that $|\mathcal{I}_{\ell,h}| \leq B$. The attention output is then approximated by re-normalizing weights over the retained set:

$$\tilde{\mathbf{o}}_{\ell,h,k} = \sum_{j \in \mathcal{I}_{\ell,h}} \tilde{A}_{\ell,h,k,j} \mathbf{v}_{\ell,h,j}. \tag{3}$$

Current policies often prioritize "heavy hitters" with the highest cumulative attention scores, assuming their dominance in preserving model performance.

# 4. Methodology

## 4.1. Long-horizon KV Cache Eviction

KV Cache eviction inherently entails a risk of information loss. Traditional eviction methods (e.g., H2O, SnapKV) rely on *instantaneous* attention weights, such as those calculated during the prefill stage. We term these methods **Heuristic Metric**. However, these methods overlook the potential for shifts in attention patterns during future decoding steps. To address this, we propose **L**ong-horizon **U**tility **KV** (**LU-KV**), a framework that evaluates KV utility over extended sequences. We formulate the cache eviction problem as a Global Combinatorial Optimization of long-horizon utility, which we solve to determine the optimal head-wise budget allocation.

**Oracle Importance.** To rigorously quantify the long-horizon marginal contribution of each token, we introduce the concept of **Oracle Importance**. Inspired by the output perturbation bound analysis in AdaKV (Feng et al., 2026b) and the criticality definition in CriticalKV (Feng et al., 2025), we define the oracle importance score $I_{\ell,h,j}$ of a cached position $j$ in head $(\ell, h)$ as its maximum potential contribution over a future decoding window:

$$I_{\ell,h,j} \triangleq \max_{k \in \{1, \ldots, K_{\max}\}} \left( A_{\ell,h,k,j} \cdot \left\| \mathbf{v}_{\ell,h,j} \mathbf{W}_O^{(\ell,h)} \right\| \right). \tag{4}$$

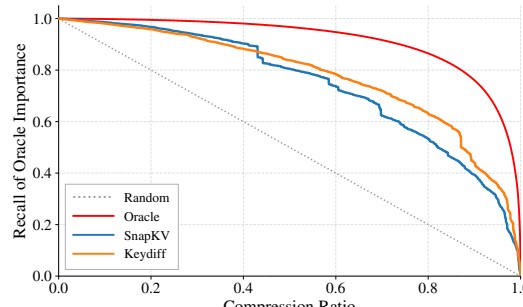

*Figure 1.* Recall of oracle importance for oracle metric and several heuristic metrics across varying compression ratios ($\sigma$) on Mistral-7B-v0.3, where 1 implies full compression and 0 implies no compression.

This score captures the true utility of a token in a long-horizon view: whether it constitutes a major component of the output vector at any future step $k$. Based on this, we theoretically construct an **Oracle Metric** ($\pi^*$) that yields the set $\mathcal{M}_{\ell,h}^{\pi^*}$, which perfectly aligns with the descending ranking of the oracle importance score $I_{\ell,h,:}$.

**Limitation of Heuristic Metric.** For a head budget $b_{\ell,h}$, $\mathcal{M}_{\ell,h}^{\pi}(b_{\ell,h})$ denotes the KV positions retained under metric $\pi$; positions outside this set are evicted. In practice, $\mathcal{M}_{\ell,h}^{\pi}(b_{\ell,h})$ determined by a heuristic metric $\pi$ often deviates from the optimal set $\mathcal{M}_{\ell,h}^{\pi^*}(b_{\ell,h})$, due to the short-horizon of $\pi$.

To analyze this discrepancy, we decompose the relationship between $\mathcal{M}_{\ell,h}^{\pi^*}$ and $\mathcal{M}_{\ell,h}^{\pi}$ into 3 classes:

- **Hits:** $\mathcal{M}_{\ell,h,\text{hit}} = \mathcal{M}_{\ell,h}^{\pi^*}(b_{\ell,h}) \cap \mathcal{M}_{\ell,h}^{\pi}(b_{\ell,h})$ (Correctly retained high oracle importance tokens)
- **Misses:** $\mathcal{M}_{\ell,h,\text{miss}} = \mathcal{M}_{\ell,h}^{\pi^*}(b_{\ell,h}) \setminus \mathcal{M}_{\ell,h}^{\pi}(b_{\ell,h})$ (High Oracle importance, wrongly evicted)
- **False Positives:** $\mathcal{M}_{\ell,h,\text{fp}} = \mathcal{M}_{\ell,h}^{\pi}(b_{\ell,h}) \setminus \mathcal{M}_{\ell,h}^{\pi^*}(b_{\ell,h})$ (Low oracle importance, wrongly retained)

Consequently, the retained cache set $\mathcal{M}_{\ell,h}^{\pi}(b_{\ell,h})$ determined by $\pi$ can also be expressed as:

$$\mathcal{M}_{\ell,h}^{\pi}(b_{\ell,h}) = (\mathcal{M}_{\ell,h}^{\pi^*}(b_{\ell,h}) \setminus \mathcal{M}_{\ell,h,\text{miss}}) \cup \mathcal{M}_{\ell,h,\text{fp}}. \tag{5}$$

Let eviction loss $\mathcal{L}_{\ell,h}(\cdot)$ denote the Oracle Importance mass lost by head $(\ell, h)$ due to removed KV-cache positions:

$$\mathcal{L}_{\ell,h}(\mathcal{M}_{\ell,h}) \triangleq \sum_{j \in \{1, \ldots, T\} \setminus \mathcal{M}_{\ell,h}} I_{\ell,h,j}. \tag{6}$$

Substituting the set formulation from Eq. (5) into the loss definition in Eq. (6), we can rigorously decompose the evic-

tion loss of policy $\pi$ into two distinct components:

$$\mathcal{L}_{\ell,h}(\mathcal{M}^{\pi}_{\ell,h}(b_{\ell,h})) = \mathcal{L}_{\ell,h}(\mathcal{M}^{\pi^*}_{\ell,h}(b_{\ell,h})) + \sum_{j \in \mathcal{M}_{\ell,h,\mathrm{miss}}} I_{\ell,h,j} - \sum_{j \in \mathcal{M}_{\ell,h,\mathrm{fp}}} I_{\ell,h,j}$$

$$= \underbrace{\mathcal{L}_{\ell,h}(\mathcal{M}^{\pi^*}_{\ell,h}(b_{\ell,h}))}_{\text{Oracle Metric Loss}} + \underbrace{\Delta_{\ell,h}(\pi, \pi^*, b_{\ell,h})}_{\text{Optimality Gap Loss}}.$$

(7)

where $\mathcal{L}_{\ell,h}(\mathcal{M}^*_{\ell,h})$ is the Oracle Metric Loss, which is fixed due to compression ratio; $\Delta_{\ell,h}(\pi, I)$ is defined as the **Optimality Gap** between the oracle metric and the used metric $\pi$ in long-horizon view, which relevant to $\pi$.

Figure 1 validates the decomposition in Eq. (7) by visualizing the total loss as the vertical distance to $Recall = 1.0$. Specifically, the loss of a heuristic metric $\mathcal{L}_{\ell,h}(\mathcal{M}^{\pi}_{\ell,h}(b_{\ell,h}))$ is the sum of the inherent oracle metric loss(red curve to 1.0) and the optimality gap loss(vertical gap between heuristic metric curve and oracle curve).

## 4.2. Global Optimization of Head-Level KV Cache Budget Allocation

The formulation above characterizes the loss within a single attention head; however, modern LLMs operate through a complex multi-head, multi-layer architecture. Existing headwise allocation approaches, e.g, AdaKV (Feng et al., 2026b), attempt to address this by employing a global greedy strategy that pools candidate tokens from all heads and retains the top-$K$ elements based on surrogate scores. Nevertheless, this strategy remains suboptimal due to the existence of the optimality gap $\Delta_{\ell,h}(\pi, \pi^*, b_{\ell,h})$ defined in Eq. 7.

**Global Optimization Objective.** We now consider the problem of allocating a global cache budget $B_{\text{total}}$ across all attention heads to minimize the aggregate eviction loss across the entire model.

Let $b_{\ell,h}$ denote the cache budget allocated to head $(\ell, h)$, subject to the global constraint $\sum_{\ell,h} b_{\ell,h} = B_{\text{total}}$. For a given metric $\pi$, we define $\mathcal{M}^{\pi}_{\ell,h}(b_{\ell,h})$ as the set of top-$b_{\ell,h}$ token positions selected by $\pi$ within that head.

The global optimization objective aims to minimize the aggregate eviction loss across all layers and heads by optimizing the budget distribution $\{b_{\ell,h}\}$:

$$\min_{\{b_{\ell,h}\}} \quad \sum_{\ell=1}^{L} \sum_{h=1}^{H} \mathcal{L}_{\ell,h}(\mathcal{M}^{\pi}_{\ell,h}(b_{\ell,h})),$$

$$\text{s.t.} \quad \sum_{\ell=1}^{L} \sum_{h=1}^{H} b_{\ell,h} = B_{\text{total}}.$$

(8)

Since the ordering induced by $\pi$ can make the discrete loss sequence $\{\mathcal{L}_{\ell,h}(\mathcal{M}^{\pi}_{\ell,h}(i))\}_{i=0}^{T}$ non-convex with respect to the integer budget, Eq. 8 forms a high-dimensional discrete combinatorial allocation problem. Although exact

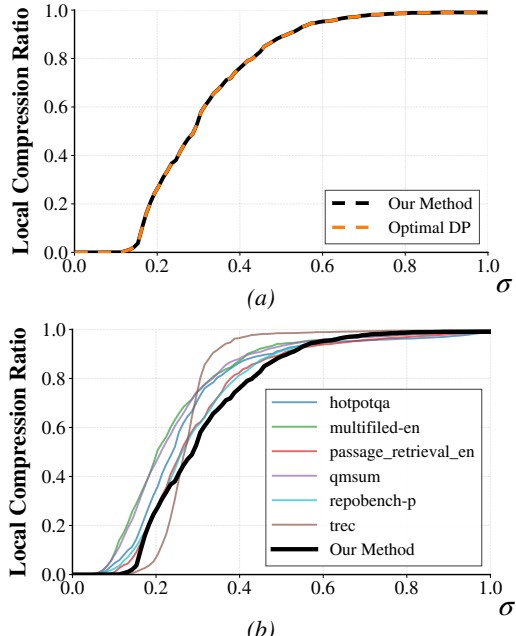

*Figure 2.* (a) Comparison between our greedy solver based on convex-hull relaxation (solving Eq. 10) and DP solution (solving Eq. 8). (b) Shows the consistent trend of optimal local compression ratio (the head-level ratio $r_{\ell,h}$) across different downstream tasks under the same global compression ratio $\sigma$. Evaluated on Mistral-7B-v0.3.

dynamic programming is possible when all per-head loss tables are available, it is costly at profiling scale. The proof of the non-convexity of $\mathcal{L}_{\ell,h}\left(\mathcal{M}^{\pi}_{\ell,h}(b_{\ell,h})\right)$ is provided in Appendix A.1.

**Efficient Optimization via Convex Hull Relaxation.** To facilitate an efficient solution to the objective in Equation 8, we propose a convex relaxation approach that transforms the discrete loss $\mathcal{L}_{\ell,h}$ into a tractable surrogate.

We first compute the raw marginal decreases of the loss sequence $\left\{\mathcal{L}_{\ell,h}\left(\mathcal{M}^{\pi}_{\ell,h}(i)\right) | 0 \le i \le T\right\}$ and apply Isotonic Regression via the Pool Adjacent Violators Algorithm (PAVA) to project them onto a non-negative, non-increasing sequence. Reconstructing the loss from these projected marginal decreases yields a convex, non-increasing surrogate sequence, denoted as $\left\{\check{\mathcal{L}}_{\ell,h}\left(\mathcal{M}^{\pi}_{\ell,h}(i)\right) | 0 \le i \le T\right\}$. We define the effective marginal gain of allocating $i$-th token in $\mathcal{M}^{\pi}_{\ell,h}$:

$$g^{\pi}_{\ell,h}(i) = \check{\mathcal{L}}_{\ell,h}\left(\mathcal{M}^{\pi}_{\ell,h}(i-1)\right) - \check{\mathcal{L}}_{\ell,h}\left(\mathcal{M}^{\pi}_{\ell,h}(i)\right) \ge 0. \quad (9)$$

The marginal gain $g^{\pi}_{\ell,h}(i)$ is monotonically non-increasing while $i$ increases. This property allows a global greedy strat-

egy to achieve the global optimum for the *relaxed* objective:

$$
\min_{\{b_{\ell,h}\}} \quad \sum_{\ell=1}^{L} \sum_{h=1}^{H} \check{\mathcal{L}}_{\ell,h}\big(\mathcal{M}_{\ell,h}^{\pi}(b_{\ell,h})\big),
$$

$$
\text{s.t.} \quad \sum_{\ell=1}^{L} \sum_{h=1}^{H} b_{\ell,h} = B_{\text{total}}. \tag{10}
$$

Specifically, we iteratively allocate the $i$-th token from the attention head $(\ell, h)$ that yields the maximum effective marginal gain $g_{\ell,h}^{\pi}(i)$, continuing until the global budget $B_{\text{total}}$ is exhausted. As illustrated in Figure 2a, our approach achieves an exact match with the results of the optimal Dynamic Programming (DP) solver. Details of the convex relaxation and allocation process are in Appendix A.2.

### 4.3. Practical Implementation: Offline Profiling

The optimization problem formulated in Eq. 10 requires future decoding results to compute the oracle importance $I$, which is inherently inaccessible during real-time inference.

However, we identify a key structural property of LLMs: individual attention heads exhibit a **consistent trend** in their optimal local-to-global compression ratios across diverse tasks. As illustrated in Figure 2b, these compression profiles remain remarkably stable across various scenarios, e.g, question answering and long-context retrieval. This empirical consistency allows us to characterize the optimal global-local budget allocation in an offline manner.

**Offline Optimal Budget Estimation.** To construct this offline allocation, we employ a data-driven probing protocol consisting of three phases:

- **Context Generation:** We construct a long-context input $C_{\text{syn}}$ ($\approx 4{,}000$ tokens) with a coherent narrative structure to simulate realistic KV cache states. In our default profile, $C_{\text{syn}}$ is an AI-generated Chinese novel excerpt and is disjoint from all evaluation benchmarks. We provide a transferability analysis of the offline profiling data in Appendix F.2.
- **Oracle Computation:** We generate a diverse set of queries $\mathcal{Q} = \{q_1, \ldots, q_M\}$ targeting different information segments. We use $M = 30$ generated questions in the default profile. For each $q_i$, the ground-truth oracle importance is computed via full-attention decoding.
- **Profile Aggregation:** We solve Eq. 10 for each query across a dense grid of global compression ratios $\rho \in [0, 1]$ to obtain the query-specific optimal local ratios $r_{\ell,h}^{*(\pi)}(q_i; \rho)$.

We aggregate these solutions into a final static profile $\Phi^{(\pi)}$ by averaging the optimal local ratios across the calibration set:

$$
\Phi^{(\pi)}(\rho)_{\ell,h} \triangleq \frac{1}{M} \sum_{i=1}^{M} r_{\ell,h}^{*(\pi)}(q_i; \rho). \tag{11}
$$

The resulting $\Phi^{(\pi)}$ serves as a lookup table mapping any target global sparsity $\rho$ to a precise configuration for every head, effectively capturing the **expected utility** of each head across the general data distribution.

**Online Execution.** During inference, our method introduces negligible computational overhead through three steps:

1. **Lookup:** Given a target global compression ratio $\sigma_{\text{target}}$, the system retrieves the pre-computed local ratios $\{r_{\ell,h}\} \leftarrow \Phi^{(\pi)}(\sigma_{\text{target}})$.
2. **Budgeting:** These ratios are translated into integer budgets: $b_{\ell,h} = \lfloor (1 - r_{\ell,h}) \cdot T \rfloor$.
3. **Eviction:** Each head independently applies the heuristic metric $\pi$ to retain the top-$b_{\ell,h}$ tokens.

This strategy successfully bridges the gap between theoretical oracle performance and practical runtime constraints without requiring online optimization.

## 5. Experiments

### 5.1. Experimental Setup

**Benchmarks.** We assess general long-context generation capabilities using **LongBench** (Bai et al., 2024), which consists of 16 diverse datasets covering various long-form tasks. Additionally, we utilize **RULER** (Hsieh et al., 2024) to evaluate retrieval robustness across expanding context windows, ranging from 4k to 128k tokens. Further details regarding the benchmarks are provided in Appendix C.

**Base Models and Baseline Methods.** We evaluate our method using LLMs of varying scales and context window capacities: Llama-3.1-8B-Instruct (Dubey et al., 2024), Mistral-7B-Instruct-v0.3 (Jiang et al., 2023), and Qwen2.5-32B-Instruct (Team, 2024). We consider two KV cache importance metrics: the **SnapKV** (Li et al., 2024) metric, denoted as $\pi_1$, which relies on accumulated attention scores; and the **KeyDiff** (Park et al., 2025) metric, denoted as $\pi_2$, which utilizes the geometric features of key vectors. Under these two metrics, we compare our approach against three allocation strategies:

- **Uniform**: A static allocation that distributes the KV budget evenly across all layers.
- **PyramidKV** (Cai et al., 2024): A static allocation based on the *Information Funneling* hypothesis, which progressively prunes the budget in deeper layers.
- **AdaKV** (Feng et al., 2026b): A dynamic allocation employing **Global Top-$k$** selection, based on the assumption

*Table 1.* **Main Results on LongBench.** Comparison of KV cache eviction strategies using the SnapKV metric ($\pi_1$) and the KeyDiff metric ($\pi_2$) at an 80% compression ratio.

| Model | Method | QA | | Summ. | Few-Shot | Synth. | Code | Avg. |
|---|---|---|---|---|---|---|---|---|
| | | Single | Multi | | | | | |
| Mistral B-v0.3 | Full-KV | 38.36 | 37.83 | 28.86 | 70.86 | 51.25 | 63.26 | 47.30 |
| | Uniform-$\pi_1$ | 24.70 | 31.78 | 24.15 | 63.95 | 47.50 | 60.98 | 40.67 |
| | Pyramid-$\pi_1$ | 25.36 | 31.95 | 23.98 | 63.35 | 48.75 | 61.80 | 40.94 |
| | Ada-$\pi_1$ | 26.21 | 31.88 | 24.42 | 66.52 | 49.50 | **62.05** | 41.89 |
| | **LU-KV-$\pi_1$ (Ours)** | **37.16** | **36.59** | **27.97** | **69.81** | **51.35** | 57.69 | **45.79** |
| | Uniform-$\pi_2$ | 28.53 | 30.86 | 24.71 | 59.00 | 33.08 | 46.89 | 36.83 |
| | Pyramid-$\pi_2$ | 29.44 | 30.53 | 24.50 | 59.33 | 34.06 | 42.95 | 36.59 |
| | Ada-$\pi_2$ | 31.94 | 33.78 | 25.23 | 59.60 | 40.79 | **57.71** | 40.54 |
| | **LU-KV-$\pi_2$ (Ours)** | **39.80** | **38.09** | **28.22** | **68.32** | **52.02** | 56.04 | **46.21** |
| Qwen 5-32B | Full-KV | 42.91 | 54.15 | 27.33 | 68.91 | 55.75 | 42.35 | 48.51 |
| | Uniform-$\pi_1$ | 25.02 | 44.50 | 23.33 | 64.45 | 48.75 | 44.99 | 41.21 |
| | Pyramid-$\pi_1$ | 19.44 | 40.23 | 21.84 | 60.24 | 50.46 | 46.34 | 38.68 |
| | Ada-$\pi_1$ | 25.75 | 43.60 | 23.34 | 65.70 | 50.13 | 44.97 | 41.58 |
| | **LU-KV-$\pi_1$ (Ours)** | **39.84** | **53.55** | **26.19** | **67.32** | **54.75** | **48.46** | **47.95** |
| | Uniform-$\pi_2$ | 26.85 | 44.30 | 22.32 | 65.82 | 38.36 | 29.32 | 38.33 |
| | Pyramid-$\pi_2$ | 22.00 | 35.87 | 20.64 | 60.05 | 24.71 | 26.85 | 32.42 |
| | Ada-$\pi_2$ | 32.16 | 46.44 | 23.68 | 66.33 | 48.92 | 36.71 | 42.32 |
| | **LU-KV-$\pi_2$ (Ours)** | **41.58** | **54.23** | **26.61** | **67.86** | **53.75** | **46.91** | **48.26** |

*Table 2.* **Main Results on RULER-16K.** Comparison of KV cache eviction strategies using the SnapKV metric ($\pi_1$) and the KeyDiff metric ($\pi_2$) at an 80% compression ratio.

| Model | Method | RULER Tasks (16K) | | | | | | | | | | | | | |
|---|---|---|---|---|---|---|---|---|---|---|---|---|---|---|---|
| | | single1 | single2 | single3 | multikey1 | multikey2 | multikey3 | multivalue | multiquery | vt | cwe | fwe | qa-1 | qa-2 | Avg |
| Mistral B-v0.3 | Full-KV | 94.20 | 96.40 | 99.60 | 97.40 | 95.60 | 76.80 | 89.50 | 88.65 | 96.28 | 82.22 | 87.93 | 71.60 | 50.00 | 86.63 |
| | Uniform-$\pi_1$ | 40.40 | 16.20 | 2.40 | 14.20 | 6.20 | 1.00 | 9.65 | 11.00 | 66.92 | 66.96 | 85.53 | 29.80 | 33.60 | 29.53 |
| | Pyramid-$\pi_1$ | 50.00 | 57.00 | 2.40 | 28.00 | 4.80 | 0.20 | 16.15 | 21.55 | 62.32 | 31.94 | 82.20 | 32.00 | 33.00 | 32.43 |
| | Ada-$\pi_1$ | 58.00 | 38.80 | 2.40 | 20.20 | 12.40 | 5.60 | 12.85 | 16.80 | 92.08 | 71.36 | **86.13** | 33.60 | 37.00 | 37.48 |
| | **LU-KV-$\pi_1$ (Ours)** | **70.80** | **78.80** | **18.20** | **83.60** | **79.20** | **67.40** | **67.80** | **76.25** | **95.88** | **78.32** | 84.47 | **62.00** | **47.00** | **69.98** |
| | Uniform-$\pi_2$ | **94.60** | 72.80 | 100.00 | 78.80 | 7.40 | 0.80 | 94.80 | 86.10 | 94.16 | 65.56 | **90.87** | 32.40 | 35.80 | 65.70 |
| | Pyramid-$\pi_2$ | 93.20 | **96.20** | 99.60 | **88.20** | 6.60 | 0.60 | 92.00 | 89.75 | **94.36** | 36.92 | 88.73 | 31.40 | 34.80 | 65.57 |
| | Ada-$\pi_2$ | 92.60 | 91.20 | 97.40 | 87.80 | 6.80 | 1.20 | 88.00 | 86.45 | 91.28 | 75.44 | 86.47 | 36.40 | 36.60 | 67.51 |
| | **LU-KV-$\pi_2$ (Ours)** | 85.60 | 76.60 | 100.00 | 87.00 | **90.80** | **35.20** | **96.45** | **92.85** | 92.16 | **80.78** | 86.80 | **64.60** | **46.80** | **79.66** |
| Qwen -32B | Full-KV | 100.00 | 100.00 | 100.00 | 100.00 | 99.80 | 100.00 | 99.85 | 99.95 | 100.00 | 97.70 | 96.20 | 79.40 | 62.40 | 95.02 |
| | Uniform-$\pi_1$ | 97.40 | 55.60 | 3.80 | 25.80 | 4.80 | 2.00 | 14.40 | 19.60 | 99.28 | 87.14 | 94.00 | 28.00 | 39.00 | 43.91 |
| | Pyramid-$\pi_1$ | 83.80 | 36.00 | 2.40 | 19.20 | 2.00 | 0.00 | 13.15 | 14.95 | 93.68 | 56.84 | **95.73** | 26.40 | 34.60 | 36.83 |
| | Ada-$\pi_1$ | 98.80 | 52.60 | 4.40 | 21.80 | 7.00 | 4.20 | 14.75 | 18.25 | 99.32 | 88.48 | 94.53 | 29.40 | 39.00 | 44.04 |
| | **LU-KV-$\pi_1$ (Ours)** | **99.80** | **99.20** | **32.00** | **84.20** | **71.80** | **78.40** | **84.60** | **85.80** | **99.72** | **95.66** | 93.13 | **65.00** | **56.80** | **80.47** |
| | Uniform-$\pi_2$ | 100.00 | 100.00 | 100.00 | 100.00 | 8.00 | 1.00 | 99.40 | 99.95 | 98.92 | 90.36 | **99.33** | 36.40 | 41.40 | 74.98 |
| | Pyramid-$\pi_2$ | 100.00 | 100.00 | 99.80 | 99.60 | 1.00 | 0.20 | **99.55** | 99.95 | 84.52 | 69.26 | 98.93 | 30.20 | 35.80 | 70.68 |
| | Ada-$\pi_2$ | 100.00 | 100.00 | 100.00 | 99.80 | 44.20 | 23.00 | 99.00 | 99.90 | 99.88 | 95.34 | 99.07 | 44.00 | 46.40 | 80.81 |
| | **LU-KV-$\pi_2$ (Ours)** | 100.00 | 100.00 | 100.00 | 100.00 | **86.60** | **46.80** | 99.25 | 99.95 | **100.00** | **96.02** | 96.20 | **71.60** | **59.00** | **88.88** |

that heads with higher importance scores warrant larger budgets.

In all experiments, **Full-KV** denotes the no-eviction upper bound retaining all KV pairs. Detailed baseline specifications are deferred to Appendix D.

**Experimental Settings.** Our evaluations are conducted using the KVPress framework (NVIDIA, 2024), adopting a *question-agnostic* compression protocol. In this setting, the context is compressed and the KV cache is evicted solely based on the input document, strictly before the arrival of any query. This paradigm closely mirrors real-world

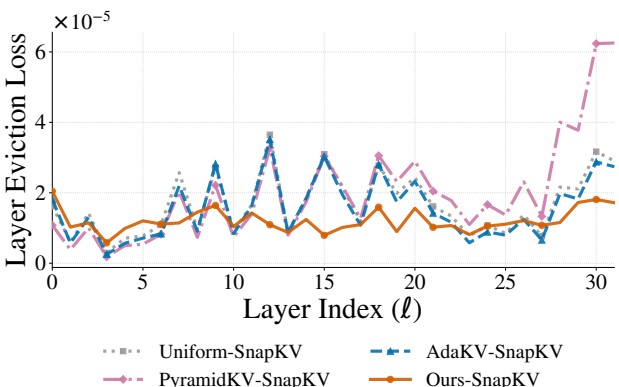

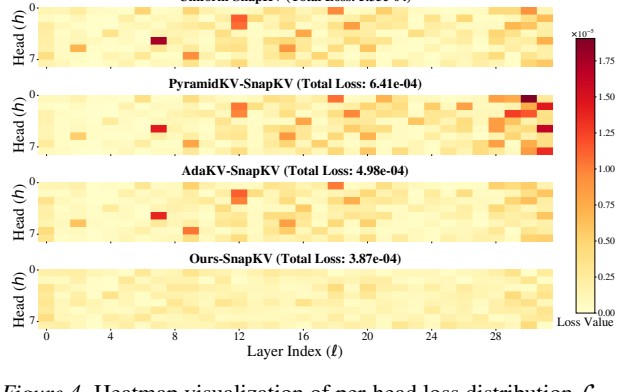

*Figure 3.* Comparison of aggregated layer-wise eviction loss. **Ours** consistently achieves the lowest and most stable loss across all layers, whereas baselines like AdaKV and PyramidKV exhibit severe loss spikes.

*Figure 4.* Heatmap visualization of per-head loss distribution $\mathcal{L}_{\ell,h}$. Baselines suffer from intense "loss bursts" (dark red blocks) in specific heads due to optimality gap, while our method effectively suppresses these spikes across the entire model.

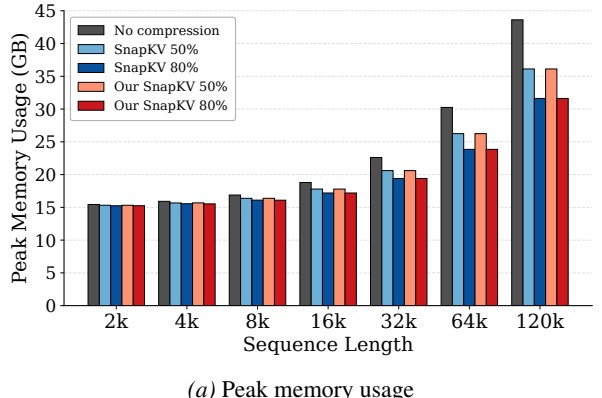

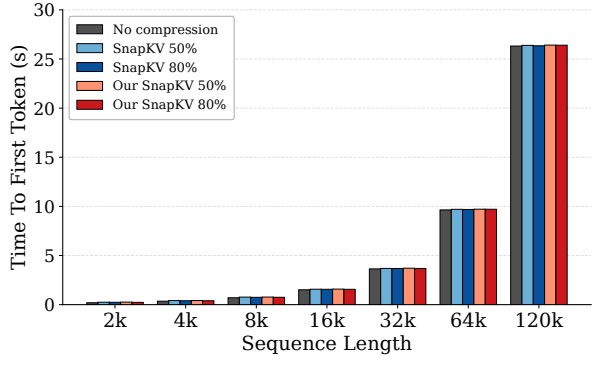

*(a)* Peak memory usage

*(b)* Time To First Token (TTFT) latency

*Figure 5.* Efficiency comparison on Llama-3.1-8b. Our method maintains comparable latency to baselines while significantly reducing memory usage in long-context scenarios.

production environments where the prompt or context is pre-filled and cached for future unknown user interactions. Furthermore, by precluding access to query-specific attention, this setup imposes a significantly more rigorous test on the method's ability to retain salient information compared to query-aware approaches.

### 5.2. Main Results on KV Cache Eviction

**Results on LongBench.** Table 1 summarizes the performance under an 80% compression ratio. Consistent with our global optimization objective in Eq. 8, our method effectively minimizes the aggregate eviction loss, translating into significant accuracy gains. On `Mistral-7B-v0.3` with $\pi_2$ (KeyDiff), our method improves the average accuracy from 40.54 (AdaKV) to 46.21, recovering 84% of the performance gap between the compressed model and the Full-KV upper bound. Crucially, these gains are robust across diverse domains, ranging from summarization to synthetic tasks, demonstrating that our learned compression profiles success-

fully capture the intrinsic *Oracle Importance* distribution across varying data densities.

**Results on RULER.** The RULER benchmark serves as a stress test for retrieval robustness in extreme contexts. Focusing on `Mistral-7B-v0.3` using the SnapKV metric ($\pi_1$) in Table 2, conventional strategies struggle significantly: Uniform allocation collapses to 29.53% average accuracy, and AdaKV provides only marginal relief at 37.48%. In contrast, our approach achieves a remarkable 69.98% average accuracy under the same 80% compression ratio. Notably, on the challenging `multikey-3` task, our method boosts performance from 1.00% (Uniform) to 67.40%, demonstrating substantial robustness in preserving sparse yet critical information.

### 5.3. LU-KV Achieves Optimal Global Allocation

To validate our core hypothesis, we visualize the eviction loss distribution under an 80% compression ratio. As il-

*Table 3.* **Ablation Study on Optimality Gap Loss.** We evaluate whether explicitly considering the optimality gap loss ($\triangle$) improves budget allocation, comparing variants w/o $\triangle$ and w/ $\triangle$. Evaluated on Mistral-7B-v0.3 and Qwen2.5-32B at an 80% compression ratio.

| Model | Method | Single-Doc QA | | | Multi-Doc QA | | | Summarization | | | Few-shot | | | Synthetic | | Code | | Avg |
|---|---|---|---|---|---|---|---|---|---|---|---|---|---|---|---|---|---|---|
| | | NrtvQA | Qasper | MF-en | Hotpot | 2WikiQA | Musique | GovRep | QMSum | MultiNews | TREC | TriviaQA | SAMSum | PCount | PR-en | Lcc | RB-P | |
| **Mistral-7B** | Full-KV | 27.04 | 38.30 | 49.75 | 49.11 | 36.68 | 27.69 | 34.64 | 25.55 | 26.40 | 76.50 | 88.96 | 47.11 | 5.50 | 97.00 | 65.60 | 60.92 | 47.30 |
| | LU-KV-$\pi_1$ (w/o $\triangle$) | 23.03 | 27.14 | 42.68 | **49.69** | 31.27 | 21.20 | 30.34 | 23.15 | 23.64 | 66.00 | 89.36 | **47.13** | 5.00 | 96.50 | **66.72** | 60.33 | 43.95 |
| | LU-KV-$\pi_1$ (w/ $\triangle$) | 25.25 | 34.91 | 51.32 | 48.87 | **38.10** | 22.80 | 33.57 | 25.02 | 25.31 | 71.00 | **91.32** | 47.12 | **5.19** | 97.50 | 53.76 | **61.62** | 45.79 |
| | LU-KV-$\pi_2$ (w/o $\triangle$) | 24.82 | 29.97 | 47.07 | 44.72 | 34.68 | 23.10 | 30.62 | 24.15 | 24.53 | 47.50 | **89.06** | 46.80 | 4.68 | 92.50 | 46.63 | 60.89 | 41.98 |
| | LU-KV-$\pi_2$ (w/ $\triangle$) | 25.80 | 39.78 | 53.82 | 48.26 | 41.33 | 24.69 | 33.49 | 25.52 | 25.65 | 69.00 | 88.81 | 47.14 | 6.53 | 97.50 | 51.18 | 60.89 | 46.21 |
| **Qwen2.5-32B** | Full-KV | 30.68 | 45.93 | 52.13 | 63.00 | 60.75 | 38.71 | 32.43 | 24.51 | 25.06 | 72.00 | 88.71 | 46.01 | 11.50 | 100.00 | 50.72 | 33.98 | 48.51 |
| | LU-KV-$\pi_1$ (w/o $\triangle$) | 27.50 | 26.07 | 36.90 | 59.58 | 53.91 | 37.27 | 29.37 | 20.76 | 22.32 | 65.50 | 88.57 | **45.45** | 9.50 | 99.25 | **60.36** | **37.77** | 45.00 |
| | LU-KV-$\pi_1$ (w/ $\triangle$) | 29.41 | 39.16 | 50.95 | 62.82 | 58.00 | 39.84 | 31.34 | 23.12 | 24.10 | 71.00 | 88.89 | 42.07 | 9.50 | **100.00** | 60.21 | 36.71 | **47.95** |
| | LU-KV-$\pi_2$ (w/o $\triangle$) | 26.65 | 26.11 | 42.18 | 57.12 | 52.08 | 30.64 | 28.25 | 22.33 | 21.16 | 73.50 | **88.69** | 43.64 | **8.50** | 91.54 | 42.14 | 36.31 | 43.18 |
| | LU-KV-$\pi_2$ (w/ $\triangle$) | **31.30** | **42.88** | 50.55 | 61.61 | 59.67 | 41.41 | 31.56 | 24.01 | 24.25 | 74.00 | 88.31 | 41.28 | 7.50 | **100.00** | 55.45 | **38.37** | **48.26** |

lustrated in Figures 3 and 4, the failure modes of existing strategies reveal fundamental limitations in both optimization granularity and deployment constraints.

**Limitations of PyramidKV.** PyramidKV optimizes along the layer-wise dimension based on fixed structural priors. While it adjusts the budget distribution across layers, Figure 3 (pink line) shows that this rigid heuristic induces a sharp escalation in loss within deeper layers (layers 27–32), failing to adapt to the high semantic density of these regions. Consequently, the aggressive pruning in deep layers—based on an ill-suited pyramidal hypothesis—causes irreversible context loss that outweighs the minor gains in shallow layers, ultimately yielding a higher aggregated Oracle Loss than the Uniform baseline.

**Limitations of AdaKV.** AdaKV focuses on the head-wise dimension, allocating budgets dynamically based on proxy score magnitudes. However, it faces a critical engineering trade-off: performing a true global cross-layer sort requires buffering all KV states during prefill, which causes unacceptable peak memory spikes. Consequently, AdaKV is often practically constrained to layer-wise uniform (or locally dynamic) budgets while competing only within layers. This explains why its layer-wise loss curve (Figure 3, blue dashed line) closely mirrors the Uniform baseline, failing to rebalance resources across layers. Furthermore, within layers, Figure 4 reveals that distinct "loss bursts" (dark red blocks) persist. This confirms the ranking discordance: simply prioritizing heads with high proxy scores fails to capture true Oracle importance, leading to suboptimal intra-layer allocation.

**Superiority of LU-KV via Optimality-Gap-Aware Allocation.** LU-KV integrates the advantages of both dimensions while circumventing their respective drawbacks. As an offline static strategy, our method retrieves the optimal con-figuration from a pre-computed profile. This allows us to execute true global optimization (cross-layer and cross-head) without incurring the runtime memory overhead that limits online dynamic methods. Results show distinct improvements in two aspects: (1) Cross-Layer: Figure 3 (orange solid line) shows that our method effectively homogenizes the eviction loss across all layers, preventing the surge in deeper layers observed in PyramidKV. (2) Cross-Head: Figure 4 confirms that the localized "loss bursts" characteristic of AdaKV are successfully eliminated. By globally optimizing the Effective Marginal Gain ($g^\pi_{\ell,h}$), LU-KV achieves superior resource utilization, significantly reducing the total Oracle Eviction Loss relative to AdaKV ($3.87 \times 10^{-4}$ vs. $4.98 \times 10^{-4}$).

### 5.4. Ablation Study

We conduct an ablation study on the optimality gap loss (Eq. 7) to assess its contribution to LU-KV. As shown in Table 3, removing the optimality gap allocation leads to substantial performance degradation across various metrics and base models. This result validates our theoretical derivation: the proxy metric for allocation is inherently suboptimal due to its deviation from the oracle importance (*i.e.*, the maximum potential contribution of a cached token over a future decoding window), and explicitly incorporating this gap into the optimization objective effectively improves KV cache allocation.

### 5.5. Efficiency Analysis

Figure 5 presents an efficiency comparison in terms of peak memory usage and Time To First Token (TTFT) latency. LU-KV achieves peak memory reduction and TTFT latency comparable to the SnapKV baseline, confirming that our method introduces negligible computational overhead. This

demonstrates that LU-KV effectively operates within strict resource constraints while significantly mitigating the performance degradation caused by KV cache compression.

## 6. Conclusion

In this paper, we theoretically analyze the limitations of existing heuristic KV eviction methods through the lens of long-horizon inference, revealing their inability to capture the long-term cumulative contribution of tokens. To bridge this gap, we introduce a novel paradigm: KV cache retention should be determined not only by instantaneous importance but also by future utility. We formulate the head-level budget allocation as a global combinatorial optimization problem and propose an efficient convex-hull relaxation and a greedy solver algorithm to solve it. Extensive evaluation across highly demanding benchmarks, such as LongBench and RULER, demonstrates the efficacy of our proposed approach.

## Acknowledgement

This work was supported by the Baige AI Team, Baidu Inc., through the Pinecone University Collaboration Program (Grant No. ZPJ2025001271-RPT1), and by the National Natural Science Foundation of China (NSFC) under Grant No. 62522206.

## Impact Statement

This work aims to make long-context LLM inference more memory- and latency-efficient by improving KV-cache budget allocation. More efficient cache use can lower serving cost and enable longer-context applications on constrained hardware. The main deployment risk is that aggressive compression can discard rare but important context, especially under domain shifts, context lengths, or decoding settings that differ from the profiling setup. Practitioners should validate the chosen compression ratio and profiling data for their target workload, and should avoid using cache compression as a substitute for safety, privacy, or reliability checks in downstream systems.

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

A. Constructing a multi-hop QA dataset for comprehen-
sive evaluation of reasoning steps. In Scott, D., Bel,
N., and Zong, C. (eds.), *Proceedings of the 28th In-
ternational Conference on Computational Linguistics*,
pp. 6609–6625, Barcelona, Spain (Online), December
2020. International Committee on Computational Lin-
guistics. doi: 10.18653/v1/2020.coling-main.580. URL
https://aclanthology.org/2020.coling
-main.580/.

Hsieh, C.-P., Sun, S., Kriman, S., Acharya, S., Rekesh, D.,
Jia, F., and Ginsburg, B. RULER: What's the real context
size of your long-context language models? In *First
Conference on Language Modeling*, 2024. URL https:
//openreview.net/forum?id=kIoBbc76Sy.

Huang, L., Cao, S., Parulian, N., Ji, H., and Wang,
L. Efficient attentions for long document summariza-
tion. In Toutanova, K., Rumshisky, A., Zettlemoyer,
L., Hakkani-Tur, D., Beltagy, I., Bethard, S., Cotterell,
R., Chakraborty, T., and Zhou, Y. (eds.), *Proceedings of
the 2021 Conference of the North American Chapter of
the Association for Computational Linguistics: Human
Language Technologies*, pp. 1419–1436, Online, June
2021. Association for Computational Linguistics. doi:
10.18653/v1/2021.naacl-main.112. URL https://ac
lanthology.org/2021.naacl-main.112/.

Jiang, A. Q., Sablayrolles, A., Mensch, A., Bamford, C.,
Chaplot, D. S., Casas, D. d. l., Bressand, F., Lengyel, G.,
Lample, G., Saulnier, L., et al. Mistral 7b. *arXiv preprint
arXiv:2310.06825*, 2023.

Joshi, M., Choi, E., Weld, D., and Zettlemoyer, L. TriviaQA:
A large scale distantly supervised challenge dataset for
reading comprehension. In Barzilay, R. and Kan, M.-Y.
(eds.), *Proceedings of the 55th Annual Meeting of the
Association for Computational Linguistics (Volume 1:
Long Papers)*, pp. 1601–1611, Vancouver, Canada, July
2017. Association for Computational Linguistics. doi:
10.18653/v1/P17-1147. URL https://aclantho
logy.org/P17-1147/.

Kim, J.-H., Kim, J., Kwon, S., Lee, J. W., Yun, S., and Song,
H. O. KVzip: Query-agnostic KV cache compression
with context reconstruction. In *The Thirty-ninth Annual
Conference on Neural Information Processing Systems*,
2026. URL https://openreview.net/forum
?id=JFygzwx8SJ.

Kočiský, T., Schwarz, J., Blunsom, P., Dyer, C., Hermann,
K. M., Melis, G., and Grefenstette, E. The NarrativeQA
reading comprehension challenge. *Transactions of the
Association for Computational Linguistics*, 6:317–328,
2018. doi: 10.1162/tacl_a_00023. URL https://ac
lanthology.org/Q18-1023/.

Li, X. and Roth, D. Learning question classifiers. In *COL-
ING 2002: The 19th International Conference on Com-
putational Linguistics*, 2002. URL https://aclant
hology.org/C02-1150/.

Li, Y., Huang, Y., Yang, B., Venkitesh, B., Locatelli,
A., Ye, H., Cai, T., Lewis, P., and Chen, D. Snapkv:
LLM knows what you are looking for before gener-
ation. In Globersons, A., Mackey, L., Belgrave, D.,
Fan, A., Paquet, U., Tomczak, J. M., and Zhang, C.
(eds.), *Advances in Neural Information Processing Sys-
tems 38: Annual Conference on Neural Information
Processing Systems 2024, NeurIPS 2024, Vancouver,
BC, Canada, December 10 - 15, 2024*, 2024. URL
http://papers.nips.cc/paper_files/p

aper/2024/hash/28ab418242603e0f7323e54185d19bde-Abstract-Conference.html.

Liu, T., Xu, C., and McAuley, J. Repobench: Benchmarking repository-level code auto-completion systems. In Kim, B., Yue, Y., Chaudhuri, S., Fragkiadaki, K., Khan, M., and Sun, Y. (eds.), *International Conference on Learning Representations*, volume 2024, pp. 47832–47850, 2024. URL https://proceedings.iclr.cc/paper_files/paper/2024/file/d191ba4c8923ed8fd8935b7c98658b5f-Paper-Conference.pdf.

NVIDIA. Kvpress, 2024. URL https://github.com/NVIDIA/kvpress.

Park, J., Jones, D., Morse, M., Goel, R., Lee, M., and Lott, C. Keydiff: Key similarity-based kv cache eviction for long-context llm inference in resource-constrained environments. In Belgrave, D., Zhang, C., Lin, H., Pascanu, R., Koniusz, P., Ghassemi, M., and Chen, N. (eds.), *Advances in Neural Information Processing Systems*, volume 38, pp. 5983–6019. Curran Associates, Inc., 2025. URL https://proceedings.neurips.cc/paper_files/paper/2025/file/0907335ecf28faf15be54485dbcbe70e-Paper-Conference.pdf.

Qin, Z., Cao, Y., Lin, M., Hu, W., Fan, S., Cheng, K., Lin, W., and Li, J. CAKE: cascading and adaptive KV cache eviction with layer preferences. In *The Thirteenth International Conference on Learning Representations, ICLR 2025, Singapore, April 24-28, 2025*. OpenReview.net, 2025. URL https://openreview.net/forum?id=EQgEMAD4kv.

Rajpurkar, P., Jia, R., and Liang, P. Know what you don't know: Unanswerable questions for SQuAD. In Gurevych, I. and Miyao, Y. (eds.), *Proceedings of the 56th Annual Meeting of the Association for Computational Linguistics (Volume 2: Short Papers)*, pp. 784–789, Melbourne, Australia, July 2018. Association for Computational Linguistics. doi: 10.18653/v1/P18-2124. URL https://aclanthology.org/P18-2124/.

Team, Q. Qwen2.5: A party of foundation models, September 2024. URL https://qwenlm.github.io/blog/qwen2.5/.

Trivedi, H., Balasubramanian, N., Khot, T., and Sabharwal, A. musique: Multihop questions via single-hop question composition. *Transactions of the Association for Computational Linguistics*, 10:539–554, 05 2022. ISSN 2307-387X. doi: 10.1162/tacl_a_00475. URL https://doi.org/10.1162/tacl_a_00475.

Xiao, G., Tian, Y., Chen, B., Han, S., and Lewis, M. Efficient streaming language models with attention sinks. In *The Twelfth International Conference on Learning Representations, ICLR 2024, Vienna, Austria, May 7-11, 2024*. OpenReview.net, 2024. URL https://openreview.net/forum?id=NG7sS51zVF.

Yang, Z., Qi, P., Zhang, S., Bengio, Y., Cohen, W., Salakhutdinov, R., and Manning, C. D. HotpotQA: A dataset for diverse, explainable multi-hop question answering. In Riloff, E., Chiang, D., Hockenmaier, J., and Tsujii, J. (eds.), *Proceedings of the 2018 Conference on Empirical Methods in Natural Language Processing*, pp. 2369–2380, Brussels, Belgium, October-November 2018. Association for Computational Linguistics. doi: 10.18653/v1/D18-1259. URL https://aclanthology.org/D18-1259/.

Zhang, Z., Sheng, Y., Zhou, T., Chen, T., Zheng, L., Cai, R., Song, Z., Tian, Y., Ré, C., Barrett, C. W., Wang, Z., and Chen, B. H2O: heavy-hitter oracle for efficient generative inference of large language models. In Oh, A., Naumann, T., Globerson, A., Saenko, K., Hardt, M., and Levine, S. (eds.), *Advances in Neural Information Processing Systems 36: Annual Conference on Neural Information Processing Systems 2023, NeurIPS 2023, New Orleans, LA, USA, December 10 - 16, 2023*, 2023. URL http://papers.nips.cc/paper_files/paper/2023/hash/6ceefa7b15572587b78ecfcebb2827f8-Abstract-Conference.html.

Zhong, M., Yin, D., Yu, T., Zaidi, A., Mutuma, M., Jha, R., Awadallah, A. H., Celikyilmaz, A., Liu, Y., Qiu, X., and Radev, D. QMSum: A new benchmark for query-based multi-domain meeting summarization. In Toutanova, K., Rumshisky, A., Zettlemoyer, L., Hakkani-Tur, D., Beltagy, I., Bethard, S., Cotterell, R., Chakraborty, T., and Zhou, Y. (eds.), *Proceedings of the 2021 Conference of the North American Chapter of the Association for Computational Linguistics: Human Language Technologies*, pp. 5905–5921, Online, June 2021. Association for Computational Linguistics. doi: 10.18653/v1/2021.naacl-main.472. URL https://aclanthology.org/2021.naacl-main.472/.

# A. Theoretical Proofs

## A.1. Non-convexity of Eviction Loss in Eq. 8

Fix an attention head $(\ell, h)$ and consider the discrete loss sequence $\left\{\mathcal{L}_{\ell,h}\left(\mathcal{M}_{\ell,h}^{\pi}(i)\right)\right\}_{i=0}^{T}$, where $\mathcal{M}_{\ell,h}^{\pi}(i)$ denotes the top-$i$ positions selected by $\pi$ in this head. Assume $I_{\ell,h,j} \geq 0$ for all cached positions $j$, and that ties in $\pi$ are resolved by a fixed deterministic rule. Since $\mathcal{M}_{\ell,h}^{\pi}(i-1) \subset \mathcal{M}_{\ell,h}^{\pi}(i)$ and $\left|\mathcal{M}_{\ell,h}^{\pi}(i) \setminus \mathcal{M}_{\ell,h}^{\pi}(i-1)\right| = 1$ for all $i \geq 1$, by Eq. (6) we have

$$
\begin{aligned}
\mathcal{L}_{\ell,h}\left(\mathcal{M}_{\ell,h}^{\pi}(i)\right) &= \sum_{j \notin \mathcal{M}_{\ell,h}^{\pi}(i)} I_{\ell,h,j} \\
&= \sum_{j \notin \mathcal{M}_{\ell,h}^{\pi}(i-1)} I_{\ell,h,j} - \sum_{j \in \mathcal{M}_{\ell,h}^{\pi}(i) \setminus \mathcal{M}_{\ell,h}^{\pi}(i-1)} I_{\ell,h,j} \\
&= \mathcal{L}_{\ell,h}\left(\mathcal{M}_{\ell,h}^{\pi}(i-1)\right) - \sum_{j \in \mathcal{M}_{\ell,h}^{\pi}(i) \setminus \mathcal{M}_{\ell,h}^{\pi}(i-1)} I_{\ell,h,j}.
\end{aligned}
\tag{12}
$$

Therefore, the discrete first difference is non-positive:

$$
\mathcal{L}_{\ell,h}\left(\mathcal{M}_{\ell,h}^{\pi}(i)\right) - \mathcal{L}_{\ell,h}\left(\mathcal{M}_{\ell,h}^{\pi}(i-1)\right) = - \sum_{j \in \mathcal{M}_{\ell,h}^{\pi}(i) \setminus \mathcal{M}_{\ell,h}^{\pi}(i-1)} I_{\ell,h,j} \leq 0.
\tag{13}
$$

The discrete second difference satisfies

$$
\begin{aligned}
&\mathcal{L}_{\ell,h}\left(\mathcal{M}_{\ell,h}^{\pi}(i+1)\right) - 2\mathcal{L}_{\ell,h}\left(\mathcal{M}_{\ell,h}^{\pi}(i)\right) + \mathcal{L}_{\ell,h}\left(\mathcal{M}_{\ell,h}^{\pi}(i-1)\right) \\
&= \left(\mathcal{L}_{\ell,h}\left(\mathcal{M}_{\ell,h}^{\pi}(i+1)\right) - \mathcal{L}_{\ell,h}\left(\mathcal{M}_{\ell,h}^{\pi}(i)\right)\right) - \left(\mathcal{L}_{\ell,h}\left(\mathcal{M}_{\ell,h}^{\pi}(i)\right) - \mathcal{L}_{\ell,h}\left(\mathcal{M}_{\ell,h}^{\pi}(i-1)\right)\right) \\
&= \sum_{j \in \mathcal{M}_{\ell,h}^{\pi}(i) \setminus \mathcal{M}_{\ell,h}^{\pi}(i-1)} I_{\ell,h,j} - \sum_{j \in \mathcal{M}_{\ell,h}^{\pi}(i+1) \setminus \mathcal{M}_{\ell,h}^{\pi}(i)} I_{\ell,h,j}.
\end{aligned}
\tag{14}
$$

Hence, $\mathcal{L}_{\ell,h}\left(\mathcal{M}_{\ell,h}^{\pi}(i)\right)$ is (discretely) convex in $i$ if and only if the oracle importances are non-increasing along the ordering induced by $\pi$. For a heuristic metric $\pi$, this condition is not guaranteed. Whenever the ordering induced by $\pi$ contains an adjacent inversion with respect to $I_{\ell,h,:}$, the right-hand side of Eq. (14) becomes negative for that $i$. Consequently, the raw loss sequence $\mathcal{L}_{\ell,h}\left(\mathcal{M}_{\ell,h}^{\pi}(b_{\ell,h})\right)$ is non-convex in general, and Eq. (8) is a non-convex discrete combinatorial allocation problem.

## A.2. Convex Relaxation Optimization of Equation 8

Eq. (8) is a discrete multi-head budget allocation problem. As shown in Appendix A.1, for a heuristic metric $\pi$, $\mathcal{L}_{\ell,h}\left(\mathcal{M}_{\ell,h}^{\pi}(b_{\ell,h})\right)$ is generally non-convex in $b_{\ell,h}$. We adopt the convex-hull relaxation described in Section 4.2 to obtain a tractable surrogate objective.

**Convex surrogate loss by PAVA.** For each head $(\ell, h)$, consider the raw discrete loss sequence $\left\{\mathcal{L}_{\ell,h}\left(\mathcal{M}_{\ell,h}^{\pi}(i)\right)\right\}_{i=0}^{T}$. We first compute its raw marginal decreases and apply isotonic regression via PAVA to project them onto a non-negative, non-increasing sequence. Reconstructing the loss by cumulative subtraction then yields a convex, non-increasing surrogate sequence $\left\{\breve{\mathcal{L}}_{\ell,h}\left(\mathcal{M}_{\ell,h}^{\pi}(i)\right)\right\}_{i=0}^{T}$, as defined in Section 4.2. We further define the effective marginal gain (Eq. (9)) as

$$
g_{\ell,h}^{\pi}(i) = \breve{\mathcal{L}}_{\ell,h}\left(\mathcal{M}_{\ell,h}^{\pi}(i-1)\right) - \breve{\mathcal{L}}_{\ell,h}\left(\mathcal{M}_{\ell,h}^{\pi}(i)\right) \geq 0,
\tag{15}
$$

which is monotonically non-increasing in $i$.

**Equivalent maximization form.** By telescoping Eq. (15), for any integer budget $b_{\ell,h} \in \{0, 1, \ldots, T\}$ we have

$$
\breve{\mathcal{L}}_{\ell,h}\left(\mathcal{M}_{\ell,h}^{\pi}(b_{\ell,h})\right) = \breve{\mathcal{L}}_{\ell,h}\left(\mathcal{M}_{\ell,h}^{\pi}(0)\right) - \sum_{i=1}^{b_{\ell,h}} g_{\ell,h}^{\pi}(i).
\tag{16}
$$

Substituting Eq. (16) into Eq. (10), the relaxed minimization is equivalent to the following maximization:

$$\max_{\{b_{\ell,h}\}} \sum_{\ell=1}^{L} \sum_{h=1}^{H} \sum_{i=1}^{b_{\ell,h}} g_{\ell,h}^{\pi}(i) \quad \text{s.t.} \quad \sum_{\ell=1}^{L} \sum_{h=1}^{H} b_{\ell,h} = B_{\text{total}}. \tag{17}$$

**Optimality of greedy allocation.** Since $g_{\ell,h}^{\pi}(i)$ is non-increasing in $i$ for every head, Eq. (17) is a separable diminishing-returns allocation problem. An optimal solution is obtained by iteratively allocating one unit of budget to the head $(\ell, h)$ that maximizes the next available gain $g_{\ell,h}^{\pi}(b_{\ell,h} + 1)$, until the budget constraint is met. Equivalently, the greedy procedure selects the $B_{\text{total}}$ largest feasible marginal gains across all heads under the prefix constraint induced by $\{g_{\ell,h}^{\pi}(i)\}_i$, which yields the global optimum of Eq. (10).

## B. Additional Experimental Results

### B.1. Visualizing the Eviction Loss across Different Metrics

To evaluate the universality of the *optimality gap*, we visualize the eviction loss for the Mistral-7B-v0.3 model on the HotpotQA task (LongBench) under an 80% global compression ratio. Figure 6 illustrates the results across three heuristic metrics: SnapKV, KeyDiff, and EA.

### B.2. Comprehensive Head-wise Optimal Allocation Profiles

In this section, we provide the full visualization of the optimal budget allocation profiles for the *Mistral-7B-v0.3* model using the KeyDiff metric. These figures display the mapping from the target global compression ratio to the allocated local compression ratio for each of the 32 layers and 8 heads.

The elements in the visualizations are defined as follows:

- The horizontal axis ($x$-axis) represents the Global Compression Ratio ($\sigma \in [0, 1]$).

- The vertical axis ($y$-axis) represents the Optimal Local Compression Ratio ($r_{\ell,h}$) for the specific head.

- The black dashed line (labeled as 'Our Method') indicates the allocation curve derived from our proposed convex-hull optimization algorithm, calculated using our synthetic calibration data.

- The orange dashed line (labeled as 'Optimal DP') represents the theoretical optimal solution computed via the Multi-Choice Knapsack Problem (MCKP)dynamic programming algorithm on the synthetic calibration data.

- The background colored lines represent the corresponding utility profiles for various downstream datasets in the benchmark.

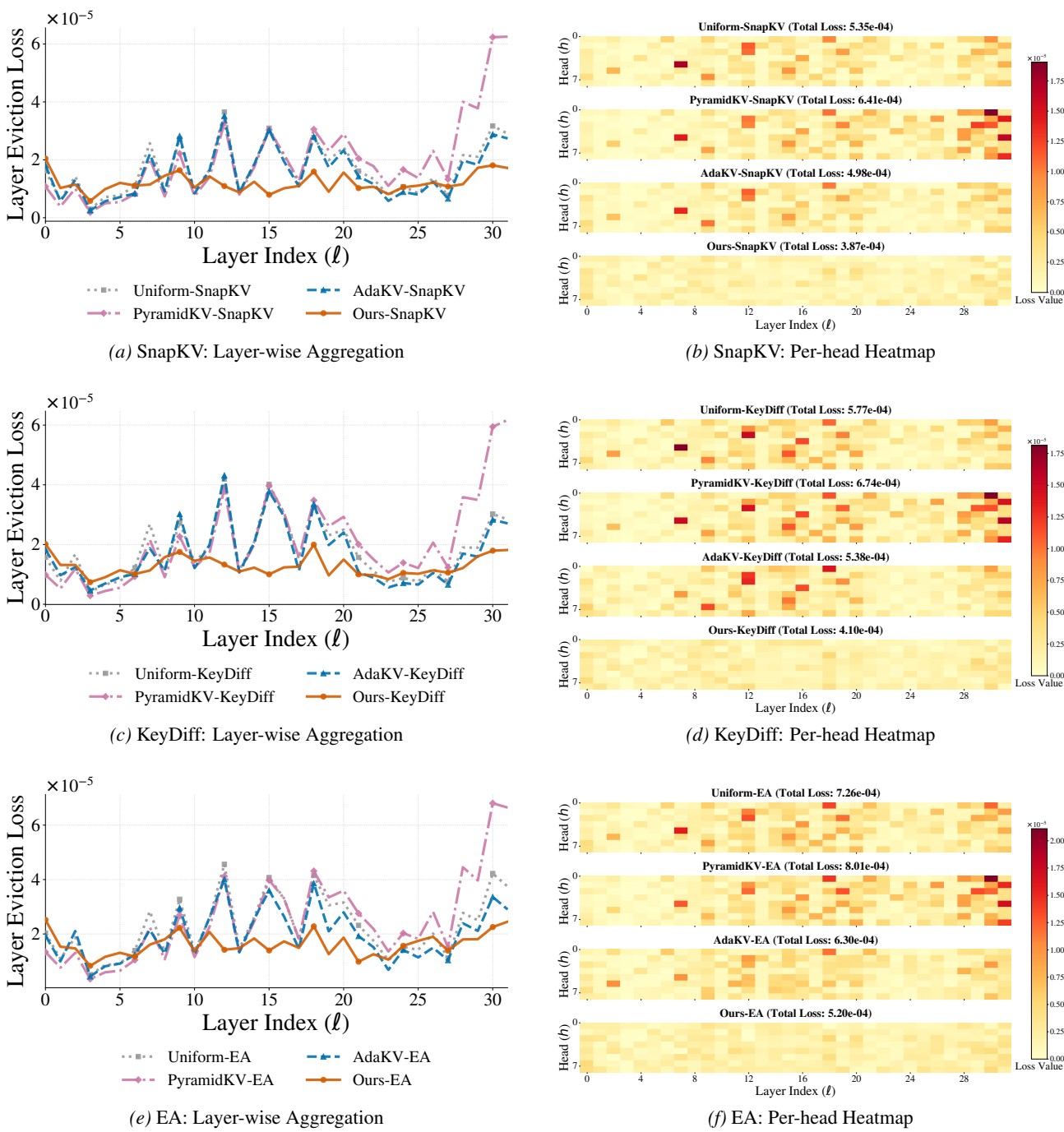

*Figure 6.* **Performance of Mistral-7B-v0.3 on HotpotQA (LongBench) across different metrics.** The figure compares the aggregated layer-wise eviction loss (left column) and per-head loss distribution heatmaps (right column) for **SnapKV** (top row), **KeyDiff** (middle row), and **EA** (bottom row) at an 80% global compression ratio.

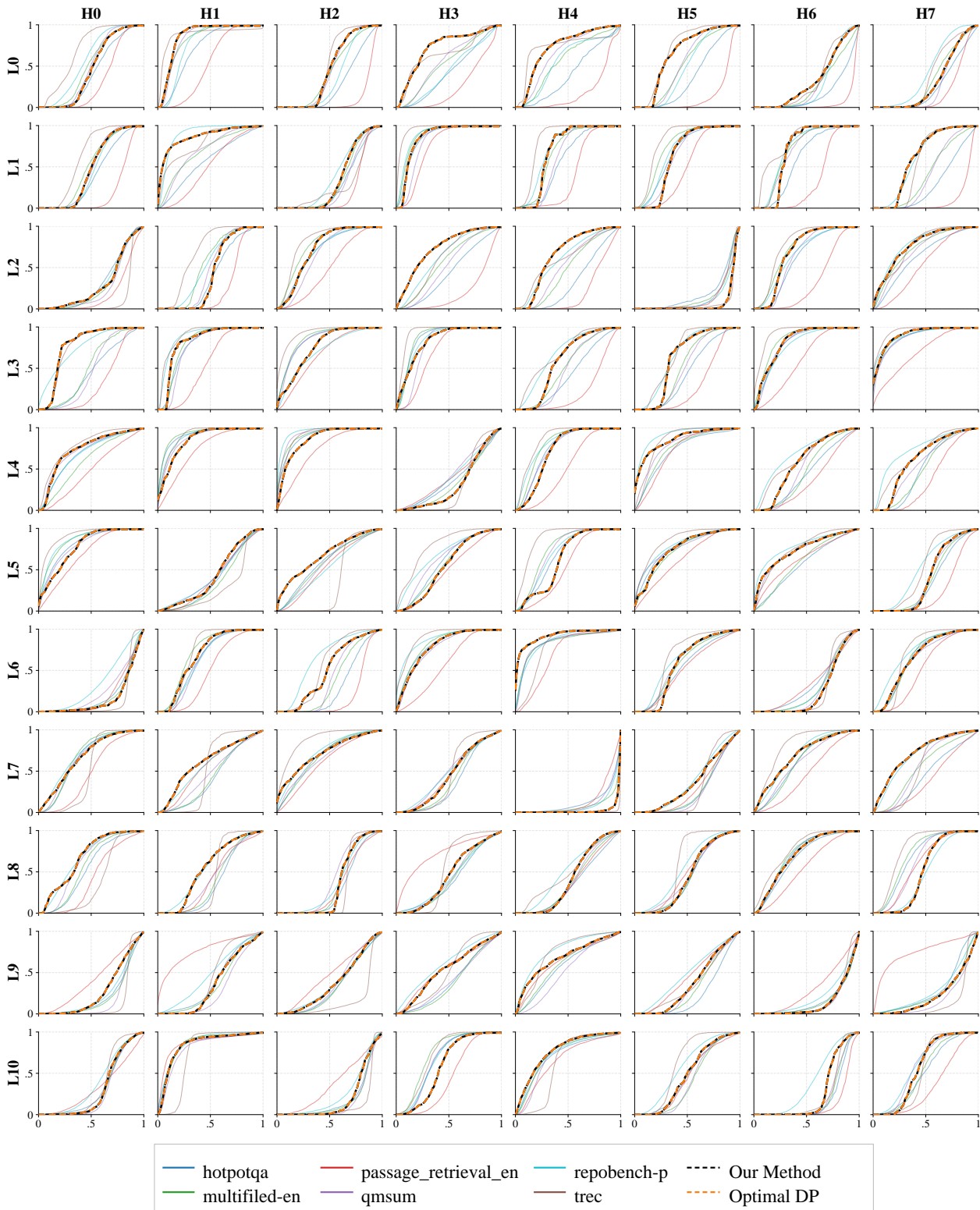

*Figure 7.* **Head-wise Optimal Allocation Profiles (Part I: Layers 0 to 10).** Visualization of the optimal local budget distribution for the **Mistral-7B-v0.3** model on the different tasks (LongBench) using the **KeyDiff** metric.

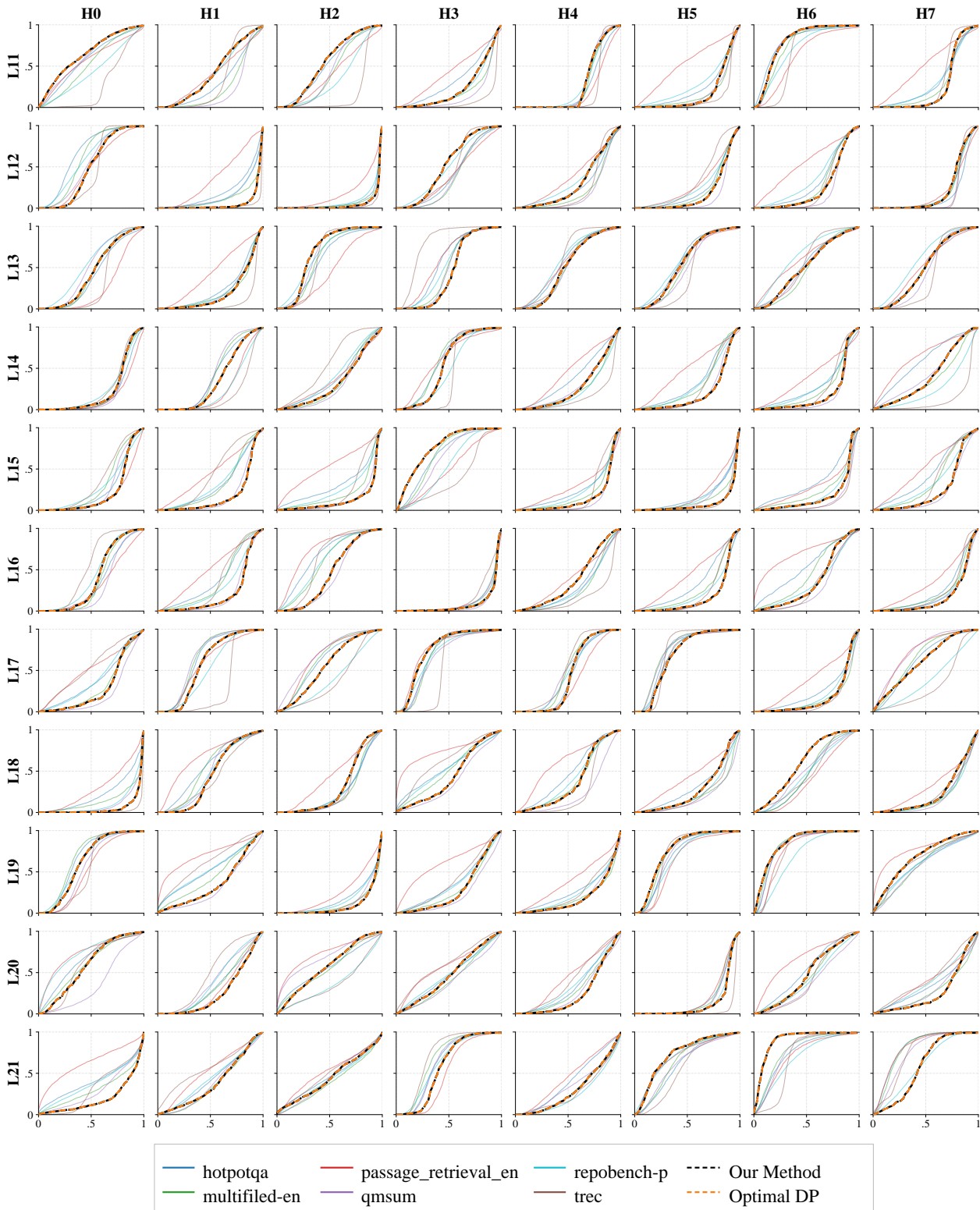

*Figure 8.* **Head-wise Optimal Allocation Profiles (Part II: Layers 11 to 21).** Visualization of the optimal local budget distribution for the **Mistral-7B-v0.3** model on the different tasks (LongBench) using the **KeyDiff** metric.

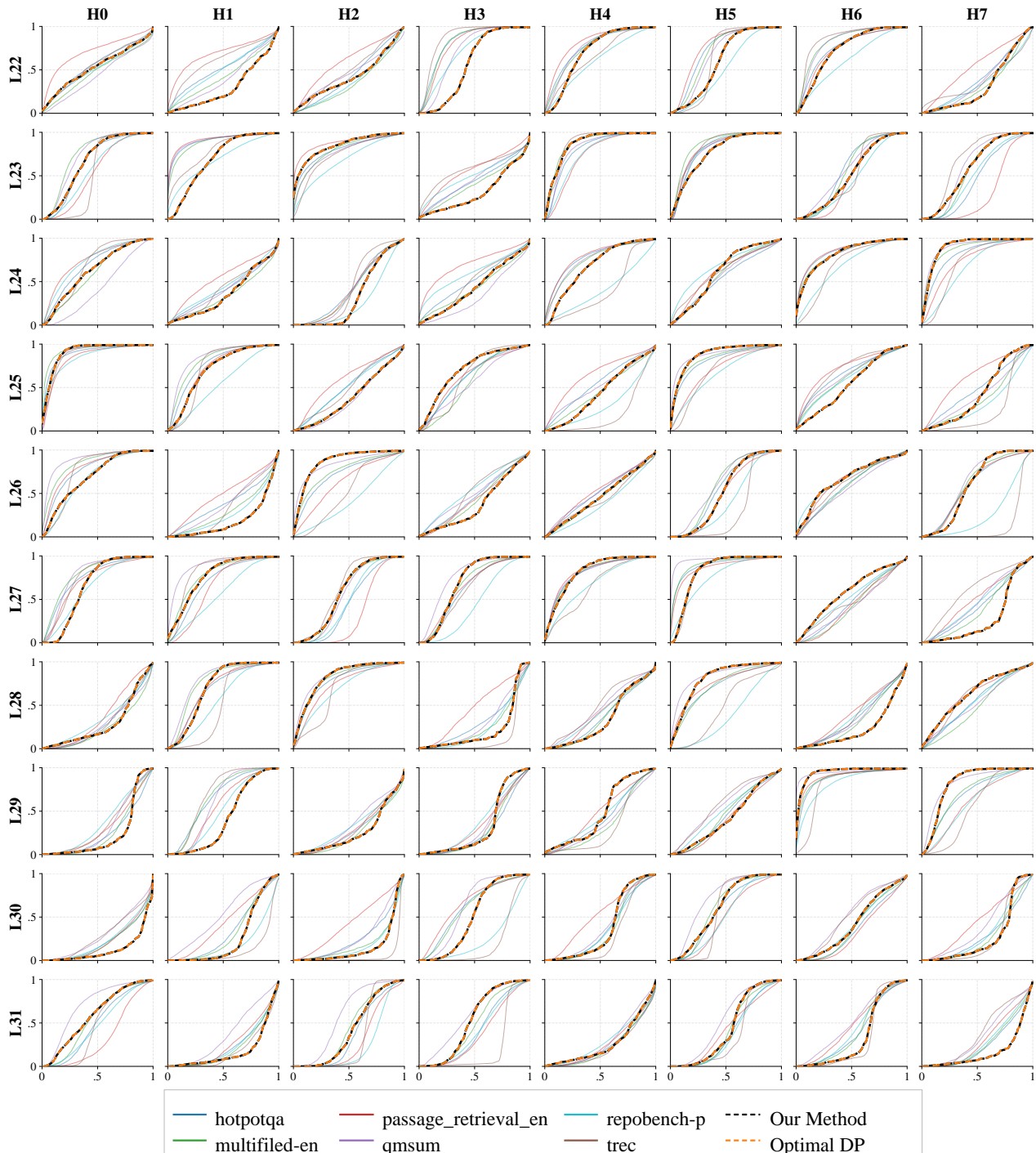

*Figure 9.* **Head-wise Optimal Allocation Profiles (Part III: Layers 22 to 31).** Visualization of the optimal local budget distribution for the **Mistral-7B-v0.3** model on the different tasks (LongBench) using the **KeyDiff** metric.

*Table 4.* Detailed scores of 16 datasets on LongBench at an 50% compression ratio.

| Model | Method | Single-Doc QA | | | Multi-Doc QA | | | Summarization | | | Few-shot | | | Synthetic | | Code | | Avg |
|---|---|---|---|---|---|---|---|---|---|---|---|---|---|---|---|---|---|---|
| | | NrtvQA | Qasper | MF-en | Hotpot | 2WikiQA | Musique | GovRep | QMSum | MultiNews | TREC | TriviaQA | SAMSum | PCount | PR-en | Lcc | RB-P | |
| **Mistral-7B-v0.3** | Full-KV | 27.04 | 38.30 | 49.75 | 49.11 | 36.68 | 27.69 | 34.64 | 25.55 | 26.40 | 76.50 | 88.96 | 47.11 | 5.50 | 97.00 | 65.60 | 60.92 | 47.30 |
| | *Metric SnapKV ($\pi_1$)* | | | | | | | | | | | | | | | | | |
| | Uniform-$\pi_1$ | 24.51 | 32.14 | 42.98 | 48.71 | 34.72 | 24.64 | 32.07 | 23.68 | 25.10 | 68.50 | 88.91 | 47.18 | 5.50 | 96.50 | 65.36 | 60.51 | 45.06 |
| | Pyramid-$\pi_1$ | 24.40 | 30.30 | 44.34 | 48.54 | 34.10 | 24.18 | 31.74 | 24.06 | 24.43 | 67.50 | 89.21 | 46.83 | 3.50 | 97.50 | 65.63 | 60.34 | 44.79 |
| | Ada-$\pi_1$ | 24.47 | 31.50 | 43.61 | **50.00** | 35.93 | 25.43 | 31.51 | 24.27 | 25.11 | 72.00 | 88.94 | 47.29 | **6.50** | 96.50 | 65.35 | 60.87 | 45.58 |
| | **LU-KV-$\pi_1$ (Ours)** | 25.81 | 38.93 | 50.53 | 49.20 | 36.82 | 27.05 | 34.96 | 25.64 | 26.42 | 76.00 | 89.45 | 47.33 | 5.54 | 98.00 | 66.14 | 61.62 | 47.46 |
| | *Metric KeyDiff ($\pi_2$)* | | | | | | | | | | | | | | | | | |
| | Uniform-$\pi_2$ | 24.03 | 35.36 | 49.01 | 47.61 | 36.29 | 25.34 | 31.97 | 24.38 | 25.46 | 56.00 | 88.56 | 46.57 | 4.20 | 95.00 | 56.82 | 60.52 | 44.20 |
| | Pyramid-$\pi_2$ | 27.26 | 34.78 | 46.46 | 45.38 | 35.94 | 25.21 | 31.73 | 24.84 | 25.35 | 68.00 | 89.21 | 47.31 | **6.50** | 96.00 | 47.92 | 60.37 | 44.52 |
| | Ada-$\pi_2$ | 26.05 | 37.68 | **51.31** | 47.04 | 37.79 | **26.43** | 33.02 | 25.05 | 25.80 | 63.00 | 88.39 | 46.82 | 2.87 | 95.25 | **65.27** | 60.39 | 45.76 |
| | **LU-KV-$\pi_2$ (Ours)** | 27.66 | 38.91 | 50.92 | 51.30 | 39.67 | 24.42 | 34.57 | 25.48 | 26.60 | 75.00 | 89.46 | 47.61 | 3.05 | 95.75 | 63.24 | 61.14 | 47.17 |
| **Llama-3.1-8B** | Full-KV | 29.39 | 45.17 | 55.74 | 58.31 | 48.12 | 32.57 | 34.53 | 25.30 | 26.91 | 72.50 | 91.78 | 44.32 | 8.47 | 99.50 | 63.43 | 52.59 | 49.29 |
| | *Metric SnapKV ($\pi_1$)* | | | | | | | | | | | | | | | | | |
| | Uniform-$\pi_1$ | 26.46 | 39.37 | 48.32 | 56.53 | 44.99 | 30.41 | 31.75 | 23.65 | 25.30 | 62.50 | 92.31 | 44.01 | 7.00 | 99.50 | **66.01** | 53.89 | 47.00 |
| | Pyramid-$\pi_1$ | 28.13 | 33.86 | 48.94 | 55.26 | 46.16 | 31.36 | 30.80 | 24.58 | 24.23 | 60.00 | **92.53** | 44.03 | 6.59 | 99.50 | 65.98 | 54.44 | 46.65 |
| | Ada-$\pi_1$ | 29.32 | 40.23 | 51.32 | 56.01 | 44.67 | **32.38** | 31.82 | 24.23 | 25.40 | 68.50 | 91.90 | **44.19** | 7.89 | 99.50 | 64.92 | 54.31 | 47.91 |
| | **LU-KV-$\pi_1$ (Ours)** | 30.57 | 44.45 | 55.59 | 56.91 | 47.32 | 32.33 | **34.64** | 25.40 | 26.64 | 71.50 | 91.65 | 44.14 | **7.95** | 100.00 | 65.25 | **55.77** | 49.38 |
| | *Metric KeyDiff ($\pi_2$)* | | | | | | | | | | | | | | | | | |
| | Uniform-$\pi_2$ | 30.58 | 41.13 | 52.39 | 55.78 | 43.72 | 29.92 | 32.73 | 24.68 | 25.07 | 65.00 | **92.03** | 45.07 | **7.83** | 99.50 | 55.89 | 54.58 | 47.24 |
| | Pyramid-$\pi_2$ | 30.66 | 41.15 | 52.06 | **56.13** | 44.73 | 30.01 | 31.85 | 25.11 | 24.98 | 59.00 | 91.78 | **45.18** | 6.08 | 99.50 | 37.66 | 55.42 | 45.71 |
| | Ada-$\pi_2$ | **31.24** | 43.64 | 51.53 | 52.68 | **48.46** | 31.66 | 34.06 | 24.38 | 26.06 | 70.00 | 91.11 | 44.80 | 7.08 | 99.50 | 63.68 | **56.85** | 48.55 |
| | **LU-KV-$\pi_2$ (Ours)** | 29.78 | 44.14 | 54.23 | 54.67 | 44.59 | 27.43 | **34.71** | 25.35 | 26.63 | 74.00 | 91.89 | 44.52 | 5.79 | 99.50 | 63.75 | 54.52 | 48.47 |
| **Qwen2.5-32B** | Full-KV | 30.68 | 45.93 | 52.13 | 63.00 | 60.75 | 38.71 | 32.43 | 24.51 | 25.06 | 72.00 | 88.71 | 46.01 | 11.50 | 100.00 | 50.72 | 33.98 | 48.51 |
| | *Metric SnapKV ($\pi_1$)* | | | | | | | | | | | | | | | | | |
| | Uniform-$\pi_1$ | 28.40 | 36.02 | 44.58 | 62.89 | 57.71 | 37.40 | 30.95 | 22.17 | 23.92 | 67.50 | 89.02 | 45.16 | **13.00** | 99.50 | 55.52 | 34.83 | 46.79 |
| | Pyramid-$\pi_1$ | 25.81 | 22.00 | 36.10 | 57.36 | 49.10 | 33.74 | 28.95 | 21.71 | 21.41 | 58.00 | **89.03** | 45.55 | 10.00 | 100.00 | **60.03** | 35.60 | 43.40 |
| | Ada-$\pi_1$ | 27.16 | 34.67 | 43.39 | 63.10 | 57.34 | 36.58 | 30.84 | 22.02 | 23.95 | 70.50 | 88.82 | **45.67** | 11.50 | 99.75 | 54.38 | 33.63 | 46.46 |
| | **LU-KV-$\pi_1$ (Ours)** | 30.73 | 44.26 | 51.59 | 63.21 | 60.75 | 39.74 | 31.96 | 24.02 | 24.72 | 71.50 | 88.38 | 45.40 | 12.00 | 100.00 | 59.30 | **37.70** | 49.08 |
| | *Metric KeyDiff ($\pi_2$)* | | | | | | | | | | | | | | | | | |
| | Uniform-$\pi_2$ | 28.66 | 37.44 | 47.88 | 62.21 | 58.97 | 39.09 | 30.86 | 23.53 | 23.37 | **75.00** | 85.93 | **46.15** | 13.25 | 98.00 | 41.03 | 34.73 | 46.63 |
| | Pyramid-$\pi_2$ | 27.84 | 30.57 | 40.25 | 55.92 | 50.81 | 32.99 | 28.69 | 22.47 | 21.50 | 65.50 | 86.66 | 45.14 | 9.67 | 75.08 | 25.86 | **35.40** | 40.90 |
| | Ada-$\pi_2$ | 29.90 | 42.44 | 51.13 | 60.14 | 59.78 | 39.67 | 31.90 | 23.21 | 23.96 | 72.00 | 87.23 | 44.84 | **13.50** | 100.00 | 43.33 | 33.71 | 47.30 |
| | **LU-KV-$\pi_2$ (Ours)** | 31.63 | 45.78 | 51.38 | 63.37 | 63.00 | 43.18 | 32.46 | 24.69 | 25.04 | 74.00 | **89.26** | 44.47 | 10.50 | 100.00 | 53.40 | 34.51 | **49.17** |

## B.3. Detailed Scores Of LongBench

In this section, we provide a comprehensive breakdown of performance across all 16 datasets in *LongBench*. Table 4 presents the results under a 50% global compression ratio for **SnapKV** ($\pi_1$) and **KeyDiff** ($\pi_2$) metrics. To further verify the universality of our approach under more aggressive compression, we also evaluate the performance at an 80% global compression ratio in Table 5, extending our analysis to include the **EA** ($\pi_3$) metric. Across these diverse settings, different importance metrics, and various model scales, the results consistently demonstrate that our proposed method remains effective.

*Table 5.* Detailed scores of 16 datasets on LongBench at an 80% compression ratio.

| Model | Method | Single-Doc QA | | | Multi-Doc QA | | | Summarization | | | Few-shot | | | Synthetic | | Code | | Avg |
|---|---|---|---|---|---|---|---|---|---|---|---|---|---|---|---|---|---|---|
| | | NrtvQA | Qasper | MF-en | Hotpot | 2WikiQA | Musique | GovRep | QMSum | MultiNews | TREC | TriviaQA | SAMSum | PCount | PR-en | Lcc | RB-P | |
| **Mistral-7B-v0.3** | Full-KV | 27.04 | 38.30 | 49.75 | 49.11 | 36.68 | 27.69 | 34.64 | 25.55 | 26.40 | 76.50 | 88.96 | 47.11 | 5.50 | 97.00 | 65.60 | 60.92 | 47.30 |
| | *Metric SnapKV ($\pi_1$)* | | | | | | | | | | | | | | | | | |
| | Uniform-$\pi_1$ | 21.37 | 19.10 | 33.64 | 44.22 | 29.19 | 21.93 | 28.20 | 22.06 | 22.20 | 55.00 | 90.07 | 46.79 | 5.00 | 90.00 | 63.37 | 58.59 | 40.67 |
| | Pyramid-$\pi_1$ | 21.77 | 19.38 | 34.92 | 43.67 | 32.21 | 19.97 | 27.77 | 22.36 | 21.80 | 55.00 | 89.21 | 45.85 | 5.50 | 92.00 | 63.53 | 60.07 | 40.94 |
| | Ada-$\pi_1$ | 20.29 | 22.31 | 36.03 | 44.43 | 29.09 | 22.12 | 27.98 | 22.88 | 22.41 | 62.50 | 90.07 | 46.99 | 5.00 | 94.00 | **64.47** | 59.62 | 41.89 |
| | **LU-KV-$\pi_1$ (Ours)** | **25.25** | **34.91** | **51.32** | **48.87** | **38.10** | **22.80** | **33.57** | **25.02** | **25.31** | **71.00** | **91.32** | **47.12** | 5.19 | **97.50** | 53.76 | **61.62** | **45.79** |
| | *Metric KeyDiff ($\pi_2$)* | | | | | | | | | | | | | | | | | |
| | Uniform-$\pi_2$ | 22.37 | 24.66 | 38.57 | 41.68 | 32.86 | 18.04 | 28.53 | 22.48 | 23.12 | 42.00 | 88.39 | 46.61 | 3.66 | 62.50 | 34.37 | 59.40 | 36.83 |
| | Pyramid-$\pi_2$ | 22.61 | 26.74 | 38.96 | 39.81 | 34.85 | 16.92 | 28.23 | 22.71 | 22.55 | 42.00 | 89.56 | 46.42 | 5.11 | 63.00 | 26.95 | 58.94 | 36.59 |
| | Ada-$\pi_2$ | 24.70 | 29.65 | 41.47 | 44.47 | 34.48 | 22.39 | 28.78 | 23.26 | 23.65 | 42.50 | **89.89** | 46.40 | 4.57 | 77.00 | **55.96** | 59.45 | 40.54 |
| | **LU-KV-$\pi_2$ (Ours)** | **25.80** | **39.78** | **53.82** | **48.26** | **41.33** | **24.69** | **33.49** | **25.52** | **25.65** | **69.00** | 88.81 | **47.14** | **6.53** | **97.50** | 51.18 | **60.89** | **46.21** |
| | *Metric EA ($\pi_3$)* | | | | | | | | | | | | | | | | | |
| | Uniform-$\pi_3$ | 12.87 | 30.45 | 40.25 | 28.62 | 23.59 | 11.32 | 21.72 | 22.81 | 24.54 | 2.00 | 13.28 | 17.38 | **4.67** | 3.55 | 17.44 | 48.62 | 20.19 |
| | Pyramid-$\pi_3$ | 14.50 | 30.96 | 36.90 | 32.05 | 27.57 | 15.42 | 27.10 | 23.75 | 24.87 | **22.50** | 18.99 | **25.83** | 2.72 | 30.50 | 20.34 | 37.34 | 24.46 |
| | Ada-$\pi_3$ | 11.08 | 18.20 | 41.28 | 26.25 | 26.43 | 10.27 | 23.24 | 22.27 | 24.43 | 2.50 | 23.75 | 13.86 | 3.32 | 9.83 | **29.84** | 50.74 | 21.08 |
| | **LU-KV-$\pi_3$ (Ours)** | **22.65** | **36.86** | **50.81** | **47.45** | **36.18** | **26.65** | **34.27** | **25.26** | **26.78** | 20.00 | **40.74** | 21.90 | 2.23 | **97.00** | 22.37 | **54.52** | **35.35** |
| **Llama-3.1-8B** | Full-KV | 29.39 | 45.17 | 55.74 | 58.31 | 48.12 | 32.57 | 34.53 | 25.30 | 26.91 | 72.50 | 91.78 | 44.32 | 8.47 | 99.50 | 63.43 | 52.59 | 49.29 |
| | *Metric SnapKV ($\pi_1$)* | | | | | | | | | | | | | | | | | |
| | Uniform-$\pi_1$ | 28.36 | 28.10 | 34.36 | 51.76 | 33.42 | 26.02 | 27.27 | 21.95 | 22.41 | 46.50 | 91.78 | 44.15 | 5.59 | 97.50 | 66.52 | 54.26 | 42.50 |
| | Pyramid-$\pi_1$ | 25.10 | 23.46 | 34.42 | 49.99 | 37.61 | 27.21 | 26.92 | 23.10 | 21.40 | 48.00 | **91.95** | 44.42 | 6.17 | 98.00 | 64.83 | 55.15 | 42.36 |
| | Ada-$\pi_1$ | 28.27 | 28.43 | 37.38 | 53.24 | 36.21 | 27.66 | 27.46 | 23.09 | 23.06 | 56.50 | 91.76 | **44.68** | 6.01 | 98.00 | **66.88** | 55.05 | 43.98 |
| | **LU-KV-$\pi_1$ (Ours)** | **29.99** | **40.10** | **56.09** | **56.39** | **46.94** | **28.51** | **32.04** | **24.32** | **25.52** | **65.50** | 89.28 | 43.76 | **6.43** | **99.50** | 59.64 | **59.27** | **47.70** |
| | *Metric KeyDiff ($\pi_2$)* | | | | | | | | | | | | | | | | | |
| | Uniform-$\pi_2$ | 29.06 | 27.88 | 39.53 | 48.54 | 30.91 | 25.13 | 28.43 | 23.57 | 21.45 | 46.50 | 91.53 | 44.29 | **9.19** | 99.50 | 41.32 | 55.02 | 41.37 |
| | Pyramid-$\pi_2$ | **30.59** | 30.06 | 39.03 | 49.80 | 35.39 | 26.51 | 27.99 | 23.32 | 21.32 | 41.00 | **91.83** | 44.30 | 4.95 | 97.50 | 29.25 | 56.26 | 40.57 |
| | Ada-$\pi_2$ | 28.95 | 34.97 | 44.70 | 50.05 | 38.63 | **28.80** | 30.29 | 23.85 | 22.98 | 60.00 | 90.65 | **44.90** | 6.99 | 98.00 | **60.66** | 56.96 | 45.09 |
| | **LU-KV-$\pi_2$ (Ours)** | 29.57 | **42.54** | **53.43** | **51.84** | **43.02** | 25.90 | **33.60** | **24.79** | **25.29** | **67.50** | 90.30 | 44.19 | 6.57 | 99.50 | 58.90 | **57.61** | **47.16** |
| | *Metric EA ($\pi_3$)* | | | | | | | | | | | | | | | | | |
| | Uniform-$\pi_3$ | 30.50 | 37.17 | 41.61 | 48.91 | 38.89 | 25.11 | 29.55 | 23.23 | 24.99 | 47.50 | 89.85 | 42.64 | 9.21 | 91.00 | 55.59 | 53.87 | 43.10 |
| | Pyramid-$\pi_3$ | 29.40 | 35.67 | 40.71 | 48.76 | 37.02 | 24.09 | 28.79 | 23.51 | 24.79 | 42.00 | 90.26 | **43.11** | **9.36** | 88.50 | 51.80 | **58.27** | 42.25 |
| | Ada-$\pi_3$ | 29.93 | 34.50 | 40.53 | 45.49 | 33.99 | 27.80 | 26.62 | 23.42 | 23.62 | 51.00 | **91.43** | 40.88 | 6.56 | 95.00 | **63.51** | 58.24 | 43.28 |
| | **LU-KV-$\pi_3$ (Ours)** | **32.13** | **46.81** | **56.09** | **53.88** | **45.64** | **30.23** | **34.12** | **25.07** | **25.77** | **71.50** | 90.12 | 42.34 | 6.59 | **99.50** | 63.38 | 56.45 | **48.73** |
| **Qwen2.5-32B** | Full-KV | 30.68 | 45.93 | 52.13 | 63.00 | 60.75 | 38.71 | 32.43 | 24.51 | 25.06 | 72.00 | 88.71 | 46.01 | 11.50 | 100.00 | 50.72 | 33.98 | 48.51 |
| | *Metric SnapKV ($\pi_1$)* | | | | | | | | | | | | | | | | | |
| | Uniform-$\pi_1$ | 24.75 | 20.48 | 29.83 | 55.54 | 45.23 | 32.74 | 28.49 | 19.97 | 21.53 | 59.00 | 88.89 | 45.47 | 9.00 | 88.50 | 55.16 | 34.82 | 41.21 |
| | Pyramid-$\pi_1$ | 18.46 | 15.29 | 24.56 | 49.62 | 39.99 | 31.07 | 26.31 | 19.75 | 19.46 | 47.50 | 88.69 | 44.52 | 9.25 | 91.67 | 59.47 | 33.20 | 38.68 |
| | Ada-$\pi_1$ | 25.99 | 21.37 | 29.88 | 54.47 | 45.43 | 30.89 | 28.34 | 19.92 | 21.76 | 62.50 | 88.88 | **45.71** | 9.50 | 90.75 | 55.38 | 34.55 | 41.58 |
| | **LU-KV-$\pi_1$ (Ours)** | **29.41** | **39.16** | **50.95** | **62.82** | **58.00** | **39.84** | **31.34** | **23.12** | **24.10** | **71.00** | **88.89** | 42.07 | 9.50 | **100.00** | **60.21** | **36.71** | **47.95** |
| | *Metric KeyDiff ($\pi_2$)* | | | | | | | | | | | | | | | | | |
| | Uniform-$\pi_2$ | 24.92 | 19.10 | 36.53 | 53.43 | 46.28 | 33.18 | 26.63 | 20.81 | 19.53 | 67.50 | 85.67 | 44.29 | 9.13 | 67.58 | 22.60 | 36.03 | 38.33 |
| | Pyramid-$\pi_2$ | 21.03 | 13.96 | 30.96 | 43.09 | 39.51 | 25.00 | 24.38 | 20.16 | 17.38 | 52.00 | 83.96 | 44.19 | 7.00 | 42.42 | 16.10 | 37.59 | 32.42 |
| | Ada-$\pi_2$ | 26.29 | 26.76 | 43.43 | 55.98 | 48.01 | 35.34 | 28.38 | 21.74 | 20.92 | 66.00 | **88.58** | 44.40 | **11.00** | 86.83 | 42.38 | 31.03 | 42.32 |
| | **LU-KV-$\pi_2$ (Ours)** | **31.30** | **42.88** | **50.55** | **61.61** | **59.67** | **41.41** | **31.56** | **24.01** | **24.25** | **74.00** | 88.31 | 41.28 | 7.50 | **100.00** | **55.45** | **38.37** | **48.26** |
| | *Metric EA ($\pi_3$)* | | | | | | | | | | | | | | | | | |
| | Uniform-$\pi_3$ | 24.65 | 32.09 | 36.49 | 54.09 | 47.96 | 32.75 | 31.21 | 22.32 | 23.41 | 70.00 | 74.20 | **43.98** | **14.61** | 86.22 | 28.61 | 33.14 | 40.98 |
| | Pyramid-$\pi_3$ | 23.00 | 27.65 | 33.70 | 43.95 | 40.11 | 19.23 | 28.89 | 21.47 | 22.73 | 44.50 | 76.41 | 42.64 | 10.41 | 63.99 | 32.60 | 31.01 | 35.14 |
| | Ada-$\pi_3$ | 30.01 | 29.86 | 45.95 | 60.11 | 54.61 | 40.05 | 31.21 | 22.70 | 24.17 | 72.00 | 82.23 | 43.26 | 8.79 | 88.02 | 46.08 | 35.53 | 44.66 |
| | **LU-KV-$\pi_3$ (Ours)** | 30.25 | **43.34** | **50.88** | **64.80** | **58.72** | **41.60** | **33.01** | **24.74** | **24.79** | **77.00** | **85.85** | 40.32 | 10.12 | **98.25** | **57.87** | **36.06** | **48.60** |

## B.4. Detailed Scores Of RULER

In this section, we provide a detailed performance breakdown on the **RULER** benchmark, evaluating the model across both RULER-16K (Table 6) and RULER-4K (Table 7). These evaluations are conducted at a strict 80% global compression ratio and include the **EA** ($\pi_3$) metric to test the robustness of our method in extreme retrieval scenarios.

The results show that traditional baselines, such as *Uniform* and *PyramidKV*, experience significant performance degradation in complex tasks like *multikey* and *variable-tracking* (vt). While *AdaKV* provides some improvement in specific configurations, it remains sensitive to the underlying heuristic metric. This is particularly evident with the **EA** metric; for example, on *Mistral-7B-v0.3* (RULER-16K), *AdaKV* achieves only 26.28% average accuracy.

In contrast, our proposed method (**LU-KV**) consistently achieves superior results across all tasks and metrics. By effectively optimizing the budget allocation, our method significantly boosts retrieval accuracy. Notably, in the aforementioned *Mistral-EA* setting, our method improves the average accuracy to 68.60%. Similar performance gains are observed for *Llama-3.1-8B* and *Qwen2.5-32B*, confirming the effectiveness of our approach across different model scales and importance metrics.

*Table 6.* Detailed scores of 13 datasets on RULER-16K at an 80% compression ratio.

| Model | Method | RULER Tasks (16K) | | | | | | | | | | | | | |
|---|---|---|---|---|---|---|---|---|---|---|---|---|---|---|---|
| | | single1 | single2 | single3 | multikey1 | multikey2 | multikey3 | multivalue | multiquery | vt | cwe | fwe | qa-1 | qa-2 | Avg |
| **Mistral-7B-v0.3** | Full-KV | 94.20 | 96.40 | 99.60 | 97.40 | 95.60 | 76.80 | 89.50 | 88.65 | 96.28 | 82.22 | 87.93 | 71.60 | 50.00 | 86.63 |
| | *Metric SnapKV ($\pi_1$)* | | | | | | | | | | | | | | |
| | Uniform-$\pi_1$ | 40.40 | 16.20 | 2.40 | 14.20 | 6.20 | 1.00 | 9.65 | 11.00 | 66.92 | 66.96 | 85.53 | 29.80 | 33.60 | 29.53 |
| | Pyramid-$\pi_1$ | 50.00 | 57.00 | 2.40 | 28.00 | 4.80 | 0.20 | 16.15 | 21.55 | 62.32 | 31.94 | 82.20 | 32.00 | 33.00 | 32.43 |
| | Ada-$\pi_1$ | 58.00 | 38.80 | 2.40 | 20.20 | 12.40 | 5.60 | 12.85 | 16.80 | 92.08 | 71.36 | **86.13** | 33.60 | 37.00 | 37.48 |
| | **LU-KV-$\pi_1$ (Ours)** | **70.80** | **78.80** | **18.20** | **83.60** | **79.20** | **67.40** | **67.80** | **76.25** | **95.88** | **78.32** | 84.47 | **62.00** | **47.00** | **69.98** |
| | *Metric KeyDiff ($\pi_2$)* | | | | | | | | | | | | | | |
| | Uniform-$\pi_2$ | **94.60** | 72.80 | 100.00 | 78.80 | 7.40 | 0.80 | 94.80 | 86.10 | 94.16 | 65.56 | **90.87** | 32.40 | 35.80 | 65.70 |
| | Pyramid-$\pi_2$ | 93.20 | **96.20** | 99.60 | **88.20** | 6.60 | 0.60 | 92.00 | 89.75 | **94.36** | 36.92 | 88.73 | 31.40 | 34.80 | 65.57 |
| | Ada-$\pi_2$ | 92.60 | 91.20 | 97.40 | 87.80 | 6.80 | 1.20 | 88.00 | 86.45 | 91.28 | 75.44 | 86.47 | 36.40 | 36.60 | 67.51 |
| | **LU-KV-$\pi_2$ (Ours)** | 85.60 | 76.60 | 100.00 | 87.00 | **90.80** | **35.20** | **96.45** | **92.85** | 92.16 | **80.78** | 86.80 | **64.60** | **46.80** | **79.66** |
| | *Metric EA ($\pi_3$)* | | | | | | | | | | | | | | |
| | Uniform-$\pi_3$ | 19.80 | 46.60 | 0.00 | 11.80 | 0.00 | 0.00 | 30.20 | 19.85 | 37.20 | 0.38 | 62.40 | 23.00 | 18.80 | 20.77 |
| | Pyramid-$\pi_3$ | 46.00 | **51.40** | 0.00 | 20.20 | 0.00 | 0.00 | 28.25 | 17.40 | 34.04 | 29.28 | 48.93 | 32.60 | 29.20 | 25.95 |
| | Ada-$\pi_3$ | 72.00 | 26.20 | 1.80 | 13.20 | 11.00 | 2.40 | 26.70 | 15.70 | 42.12 | 0.34 | **87.13** | 23.80 | 19.20 | 26.28 |
| | **LU-KV-$\pi_3$ (Ours)** | **84.20** | 21.80 | **58.60** | **68.80** | **96.00** | **55.60** | **78.45** | **60.10** | **93.52** | **78.96** | 84.73 | **62.40** | **48.60** | **68.60** |
| **Llama-3.1-8B** | Full-KV | 100.00 | 100.00 | 100.00 | 99.60 | 100.00 | 99.20 | 98.70 | 99.10 | 99.80 | 88.80 | 89.93 | 81.20 | 57.00 | 93.33 |
| | *Metric SnapKV ($\pi_1$)* | | | | | | | | | | | | | | |
| | Uniform-$\pi_1$ | 98.00 | 83.60 | 2.60 | 52.20 | 11.20 | 4.40 | 34.90 | 40.50 | 89.40 | 14.22 | 79.00 | 28.80 | 30.80 | 43.82 |
| | Pyramid-$\pi_1$ | 90.60 | 97.80 | 2.40 | 85.40 | 20.60 | 0.80 | 72.70 | 76.85 | 84.64 | 11.10 | 80.33 | 29.40 | 33.00 | 52.74 |
| | Ada-$\pi_1$ | 99.20 | 90.20 | 3.00 | 69.20 | 20.60 | 19.20 | 47.30 | 55.20 | 95.92 | 42.54 | **86.47** | 32.40 | 33.20 | 53.42 |
| | **LU-KV-$\pi_1$ (Ours)** | **100.00** | **99.80** | **54.00** | **99.40** | **85.20** | **93.60** | **98.60** | **98.80** | **97.44** | **61.40** | 84.67 | **65.40** | **49.80** | **83.70** |
| | *Metric KeyDiff ($\pi_2$)* | | | | | | | | | | | | | | |
| | Uniform-$\pi_2$ | 100.00 | 100.00 | 100.00 | 99.60 | 16.40 | 0.00 | **99.25** | 99.55 | 98.28 | 59.70 | 86.93 | 38.40 | 43.00 | 72.39 |
| | Pyramid-$\pi_2$ | 100.00 | 100.00 | 100.00 | 99.60 | 11.00 | 0.00 | 98.90 | **99.60** | **99.44** | 23.14 | 85.93 | 39.80 | 40.60 | 69.08 |
| | Ada-$\pi_2$ | 100.00 | 100.00 | 100.00 | 99.60 | 26.00 | 0.80 | 98.55 | 99.25 | 98.60 | 78.34 | **90.93** | 45.60 | 42.00 | 75.36 |
| | **LU-KV-$\pi_2$ (Ours)** | 100.00 | 100.00 | 100.00 | 99.20 | **99.20** | **56.00** | 99.20 | 99.20 | 97.36 | **80.80** | 87.53 | **76.20** | **52.60** | **88.25** |
| | *Metric EA ($\pi_3$)* | | | | | | | | | | | | | | |
| | Uniform-$\pi_3$ | 98.60 | 96.20 | 2.40 | 85.80 | 5.20 | 0.00 | 79.45 | 89.75 | 81.00 | 17.44 | 61.33 | 51.60 | 40.20 | 54.54 |
| | Pyramid-$\pi_3$ | 98.80 | 88.80 | 1.20 | 84.80 | 10.00 | 0.00 | 76.65 | 80.65 | 86.80 | 5.88 | 36.20 | 52.00 | 41.00 | 50.98 |
| | Ada-$\pi_3$ | 99.80 | 99.40 | 3.80 | 90.60 | 38.80 | 2.60 | 75.55 | 91.60 | 95.28 | 9.56 | 81.40 | 43.20 | 40.80 | 59.41 |
| | **LU-KV-$\pi_3$ (Ours)** | **100.00** | **100.00** | **99.60** | **99.40** | **99.80** | **98.60** | **97.55** | **98.80** | **98.04** | **80.18** | **88.13** | **78.40** | **54.00** | **91.73** |
| **Qwen2.5-32B** | Full-KV | 100.00 | 100.00 | 100.00 | 100.00 | 99.80 | 100.00 | 99.85 | 99.95 | 100.00 | 97.70 | 96.20 | 79.40 | 62.40 | 95.02 |
| | *Metric SnapKV ($\pi_1$)* | | | | | | | | | | | | | | |
| | Uniform-$\pi_1$ | 97.40 | 55.60 | 3.80 | 25.80 | 4.80 | 2.00 | 14.40 | 19.60 | 99.28 | 87.14 | 94.00 | 28.00 | 39.00 | 43.91 |
| | Pyramid-$\pi_1$ | 83.80 | 36.00 | 2.40 | 19.20 | 2.00 | 0.00 | 13.15 | 14.95 | 93.68 | 56.84 | **95.73** | 26.40 | 34.60 | 36.83 |
| | Ada-$\pi_1$ | 98.80 | 52.60 | 4.40 | 21.80 | 7.00 | 4.20 | 14.75 | 18.25 | 99.32 | 88.48 | 94.53 | 29.40 | 39.00 | 44.04 |
| | **LU-KV-$\pi_1$ (Ours)** | **99.80** | **99.20** | **32.00** | **84.20** | **71.80** | **78.40** | **84.60** | **85.80** | **99.72** | **95.66** | 93.13 | **65.00** | **56.80** | **80.47** |
| | *Metric KeyDiff ($\pi_2$)* | | | | | | | | | | | | | | |
| | Uniform-$\pi_2$ | 100.00 | 100.00 | 100.00 | 100.00 | 8.00 | 1.00 | 99.40 | 99.95 | 98.92 | 90.36 | **99.33** | 36.40 | 41.40 | 74.98 |
| | Pyramid-$\pi_2$ | 100.00 | 100.00 | 99.80 | 99.60 | 1.00 | 0.20 | **99.55** | 99.95 | 84.52 | 69.26 | 98.93 | 30.20 | 35.80 | 70.68 |
| | Ada-$\pi_2$ | 100.00 | 100.00 | 100.00 | 99.80 | 44.20 | 23.00 | 99.00 | 99.90 | 99.88 | 95.34 | 99.07 | 44.00 | 46.40 | 80.81 |
| | **LU-KV-$\pi_2$ (Ours)** | 100.00 | 100.00 | 100.00 | 100.00 | **86.60** | **46.80** | 99.25 | 99.95 | **100.00** | **96.02** | 96.20 | **71.60** | **59.00** | **88.88** |
| | *Metric EA ($\pi_3$)* | | | | | | | | | | | | | | |
| | Uniform-$\pi_3$ | 93.40 | 97.60 | 6.00 | 97.40 | 0.40 | 0.00 | 94.75 | 98.75 | 99.44 | 73.10 | 80.00 | 49.20 | 48.20 | 64.48 |
| | Pyramid-$\pi_3$ | 81.80 | 69.00 | 0.20 | 55.20 | 0.00 | 0.00 | 56.30 | 61.20 | 96.12 | 10.94 | 39.93 | 39.20 | 41.80 | 42.44 |
| | Ada-$\pi_3$ | **100.00** | **100.00** | 22.00 | 99.20 | 96.40 | **39.40** | 98.40 | 99.55 | 99.88 | 92.04 | **93.93** | 60.20 | 50.80 | 80.91 |
| | **LU-KV-$\pi_3$ (Ours)** | 99.60 | 99.80 | **91.00** | **100.00** | **99.40** | 29.40 | **99.35** | **99.95** | **100.00** | **95.98** | 93.00 | **79.40** | **60.60** | **88.27** |

*Table 7.* Detailed scores of 13 datasets on RULER-4K at an 80% compression ratio.

| Model | Method | RULER Tasks (4K) | | | | | | | | | | | | | |
|---|---|---|---|---|---|---|---|---|---|---|---|---|---|---|---|
| | | single1 | single2 | single3 | multikey1 | multikey2 | multikey3 | multivalue | multiquery | vt | cwe | fwe | qa-1 | qa-2 | Avg |
| **Mistral-7B-v0.3** | Full-KV | 93.20 | 95.80 | 100.00 | 99.40 | 100.00 | 97.40 | 89.05 | 97.50 | 99.56 | 98.46 | 95.60 | 76.80 | 54.60 | 92.11 |
| | *Metric SnapKV ($\pi_1$)* | | | | | | | | | | | | | | |
| | Uniform-$\pi_1$ | 37.00 | 13.00 | 2.40 | 18.40 | 5.80 | 0.60 | 11.80 | 17.10 | 25.16 | 84.66 | 82.40 | 40.60 | 35.60 | 28.81 |
| | Pyramid-$\pi_1$ | 35.20 | 16.40 | 2.40 | 19.00 | 4.00 | 0.00 | 11.40 | 17.10 | 28.80 | 42.84 | 75.40 | 35.20 | 33.80 | 24.73 |
| | Ada-$\pi_1$ | 50.40 | 13.00 | 2.40 | 18.20 | 12.80 | 5.80 | 12.70 | 17.25 | 49.84 | 89.54 | 87.00 | 44.60 | 40.20 | 34.13 |
| | **LU-KV-$\pi_1$ (Ours)** | **79.00** | **83.60** | **12.60** | **71.20** | **89.80** | **93.40** | **54.50** | **66.80** | **93.64** | **94.80** | **93.07** | **68.40** | **49.60** | **73.11** |
| | *Metric KeyDiff ($\pi_2$)* | | | | | | | | | | | | | | |
| | Uniform-$\pi_2$ | 90.60 | 97.00 | 100.00 | 83.20 | 8.00 | 0.00 | 88.10 | 89.40 | 98.92 | 60.48 | 81.33 | 39.60 | 28.80 | 66.57 |
| | Pyramid-$\pi_2$ | **94.40** | **97.20** | 100.00 | 84.80 | 6.00 | 0.00 | 83.80 | 89.30 | 98.84 | 14.72 | 77.53 | 37.60 | 23.80 | 62.15 |
| | Ada-$\pi_2$ | 91.80 | 95.00 | 99.40 | 83.80 | 5.80 | 0.00 | 82.00 | 86.80 | 99.36 | 74.22 | 75.93 | 45.80 | 32.00 | 67.07 |
| | **LU-KV-$\pi_2$ (Ours)** | 87.40 | 96.00 | 99.40 | **98.40** | **98.80** | **50.20** | **96.35** | **97.40** | 99.36 | **92.28** | **92.93** | **72.60** | **49.20** | **86.95** |
| | *Metric EA ($\pi_3$)* | | | | | | | | | | | | | | |
| | Uniform-$\pi_3$ | 66.40 | 57.40 | 0.40 | 44.00 | 0.60 | 0.00 | 56.30 | 33.95 | 57.92 | 71.06 | 79.87 | 49.00 | 41.40 | 42.95 |
| | Pyramid-$\pi_3$ | 72.00 | 51.60 | 0.00 | 38.00 | 0.60 | 0.00 | 39.75 | 28.15 | 80.44 | 57.70 | 64.07 | 46.20 | 38.80 | 39.79 |
| | Ada-$\pi_3$ | **86.00** | 39.20 | 3.00 | 37.60 | 34.00 | 41.00 | 59.20 | 34.40 | 97.84 | 90.94 | 92.47 | 51.60 | 40.80 | 54.47 |
| | **LU-KV-$\pi_3$ (Ours)** | 83.80 | **83.20** | **98.60** | **97.20** | **99.80** | **97.00** | **82.05** | **90.15** | **99.52** | **96.50** | **95.60** | **75.60** | **55.60** | **88.82** |
| **Llama-3.1-8B** | Full-KV | 100.00 | 100.00 | 100.00 | 99.80 | 100.00 | 99.80 | 99.90 | 99.90 | 99.88 | 99.68 | 94.93 | 88.00 | 62.60 | 95.73 |
| | *Metric SnapKV ($\pi_1$)* | | | | | | | | | | | | | | |
| | Uniform-$\pi_1$ | 82.00 | 70.80 | 2.40 | 39.80 | 13.20 | 2.60 | 38.35 | 35.65 | 61.16 | 62.06 | 81.47 | 42.80 | 32.80 | 43.47 |
| | Pyramid-$\pi_1$ | 74.80 | 98.20 | 2.40 | 77.00 | 23.00 | 0.00 | 69.70 | 69.80 | 48.48 | 37.42 | 71.07 | 39.00 | 29.60 | 49.27 |
| | Ada-$\pi_1$ | 86.60 | 69.80 | 2.40 | 44.40 | 17.60 | 13.80 | 37.75 | 38.80 | 81.20 | 89.82 | 87.33 | 48.20 | 36.80 | 50.35 |
| | **LU-KV-$\pi_1$ (Ours)** | **99.40** | **99.60** | **14.20** | **99.80** | **87.20** | **95.60** | **98.30** | **99.80** | **97.52** | **96.08** | **93.53** | **77.60** | **55.40** | **85.69** |
| | *Metric KeyDiff ($\pi_2$)* | | | | | | | | | | | | | | |
| | Uniform-$\pi_2$ | 100.00 | 100.00 | 100.00 | 99.80 | 14.60 | 0.00 | 98.65 | 99.85 | 98.08 | 39.42 | 82.53 | 38.60 | 27.00 | 69.12 |
| | Pyramid-$\pi_2$ | 100.00 | 99.80 | 100.00 | **100.00** | 7.80 | 0.00 | 99.25 | 99.80 | **99.64** | 20.70 | 78.13 | 41.20 | 27.60 | 67.22 |
| | Ada-$\pi_2$ | 100.00 | 99.80 | 100.00 | 99.60 | 35.00 | 0.20 | 99.40 | 99.65 | 99.08 | 68.36 | 82.20 | 54.20 | 36.00 | 74.88 |
| | **LU-KV-$\pi_2$ (Ours)** | 100.00 | 100.00 | 100.00 | 99.80 | **99.60** | **56.80** | **99.80** | **99.90** | 99.24 | **92.50** | **93.80** | **83.20** | **57.20** | **90.91** |
| | *Metric EA ($\pi_3$)* | | | | | | | | | | | | | | |
| | Uniform-$\pi_3$ | 98.60 | 91.40 | 1.60 | 92.00 | 13.20 | 0.00 | 90.20 | 93.35 | 81.92 | 43.32 | 73.00 | 55.60 | 45.40 | 59.97 |
| | Pyramid-$\pi_3$ | 99.40 | 81.60 | 1.00 | 89.80 | 22.60 | 0.00 | 86.75 | 92.00 | 97.12 | 26.64 | 58.27 | 57.00 | 44.60 | 58.21 |
| | Ada-$\pi_3$ | 99.80 | 95.00 | 5.00 | 89.20 | 55.40 | 4.60 | 78.65 | 88.65 | 92.92 | 88.94 | 90.40 | 42.60 | 44.80 | 67.38 |
| | **LU-KV-$\pi_3$ (Ours)** | **100.00** | **99.80** | **100.00** | **100.00** | **99.80** | **99.40** | **99.90** | **99.90** | **98.60** | **97.46** | **95.07** | **83.60** | **59.60** | **94.86** |
| **Qwen2.5-32B** | Full-KV | 100.00 | 100.00 | 100.00 | 99.80 | 100.00 | 100.00 | 99.95 | 100.00 | 100.00 | 99.86 | 98.60 | 89.40 | 67.60 | 96.55 |
| | *Metric SnapKV ($\pi_1$)* | | | | | | | | | | | | | | |
| | Uniform-$\pi_1$ | **95.40** | 39.00 | 3.60 | 27.20 | 4.80 | 0.40 | 25.50 | 22.75 | 92.64 | 96.18 | 86.27 | 56.20 | 41.80 | 45.52 |
| | Pyramid-$\pi_1$ | 80.20 | 13.00 | 2.40 | 13.40 | 1.60 | 0.00 | 13.00 | 12.70 | 57.60 | 63.30 | 66.67 | 34.00 | 31.00 | 29.91 |
| | Ada-$\pi_1$ | 93.60 | 24.00 | 2.40 | 19.60 | 7.00 | 4.00 | 17.55 | 17.40 | 94.80 | 98.50 | 89.00 | 59.00 | 42.60 | 43.80 |
| | **LU-KV-$\pi_1$ (Ours)** | 93.00 | **92.20** | **16.40** | **81.40** | **80.60** | **84.20** | **91.00** | **87.90** | **99.08** | **99.68** | **95.40** | **85.00** | **63.80** | **82.28** |
| | *Metric KeyDiff ($\pi_2$)* | | | | | | | | | | | | | | |
| | Uniform-$\pi_2$ | 100.00 | 100.00 | 100.00 | 99.60 | 4.60 | 0.40 | 99.95 | 100.00 | 76.00 | 78.94 | 88.47 | 49.20 | 34.20 | 71.64 |
| | Pyramid-$\pi_2$ | 100.00 | 100.00 | 99.80 | 99.40 | 1.00 | 0.00 | 99.95 | 99.75 | 26.56 | 8.10 | 75.00 | 35.40 | 24.40 | 59.18 |
| | Ada-$\pi_2$ | 100.00 | 100.00 | 100.00 | 99.80 | 37.60 | 7.60 | 99.95 | 100.00 | 99.68 | 94.72 | 87.53 | 59.40 | 40.60 | 78.99 |
| | **LU-KV-$\pi_2$ (Ours)** | 100.00 | 100.00 | 100.00 | 99.80 | **96.20** | **39.80** | 99.95 | 100.00 | **100.00** | **99.20** | **95.53** | **87.80** | **64.20** | **90.96** |
| | *Metric EA ($\pi_3$)* | | | | | | | | | | | | | | |
| | Uniform-$\pi_3$ | 86.80 | 92.80 | 3.00 | 91.80 | 0.80 | 0.00 | 91.70 | 91.85 | 99.60 | 90.28 | 82.13 | 62.20 | 52.20 | 65.01 |
| | Pyramid-$\pi_3$ | 71.20 | 49.00 | 0.00 | 33.00 | 0.00 | 0.00 | 30.90 | 35.20 | 92.68 | 27.58 | 38.93 | 49.80 | 41.20 | 36.11 |
| | Ada-$\pi_3$ | **100.00** | 99.80 | 26.60 | 98.20 | 95.40 | **69.80** | 99.65 | 99.55 | 99.88 | 99.68 | 94.00 | 73.00 | 59.60 | 85.78 |
| | **LU-KV-$\pi_3$ (Ours)** | 99.80 | **100.00** | **97.20** | **99.60** | **100.00** | 51.00 | **99.90** | **100.00** | **100.00** | **99.86** | **96.60** | **86.00** | **64.60** | **91.89** |

# C. Details about Benchmarks

To evaluate the long-context capabilities of the models comprehensively, we employ two distinct benchmarks: RULER and LongBench. These benchmarks provide complementary insights, with RULER offering controllable synthetic stress tests and LongBench providing realistic multi-task evaluations.

## C.1. RULER

RULER (Hsieh et al., 2024) is a synthetic benchmark designed to evaluate long-context language models beyond the standard retrieval-based "needle-in-a-haystack" (NIAH) tests. Unlike simple retrieval tasks, RULER introduces flexible configurations to customize sequence length and task complexity. It categorizes tasks into four distinct domains to test behaviors beyond searching from context:

- **Retrieval:** Extending the vanilla NIAH, this category includes Single NIAH (S-NIAH), Multi-keys NIAH (MK-NIAH), Multi-values NIAH (MV-NIAH), and Multi-queries NIAH (MQ-NIAH). These tasks test the model's robustness against distractors and its ability to retrieve diverse types and quantities of needles.

- **Multi-hop Tracing:** To evaluate coreference chain resolution, RULER utilizes a Variable Tracking (VT) task, requiring the model to trace variable assignment chains across the long context.

- **Aggregation:** This category tests the ability to aggregate relevant information spanning long-range context. Tasks include Common Words Extraction (CWE) and Frequent Words Extraction (FWE), where the model identifies words based on their frequency distribution.

- **Question Answering (QA):** This domain uses augmented versions of SQuAD (Rajpurkar et al., 2018) and Hot-potQA (Yang et al., 2018) with inserted distractors to simulate long-context question answering scenarios.

## C.2. LongBench

Complementing the synthetic nature of RULER, we utilize LongBench (Bai et al., 2024), a multi-task benchmark designed to assess long-context understanding in realistic scenarios. In this work, we focus specifically on the 16 English datasets from LongBench, which cover six major task categories. The English subset comprises:

- **Single-Document QA:** Evaluated using NarrativeQA (Kočiský et al., 2018), Qasper (Dasigi et al., 2021), and MultiFieldQA-en, requiring models to comprehend long individual documents.

- **Multi-Document QA:** Involves complex reasoning across multiple documents, utilizing HotpotQA (Yang et al., 2018), 2WikiMultihopQA (Ho et al., 2020), and MuSiQue (Trivedi et al., 2022).

- **Summarization:** Tests the ability to synthesize long inputs using GovReport (Huang et al., 2021), QMSum (Zhong et al., 2021), and MultiNews (Fabbri et al., 2019).

- **Few-Shot Learning:** Assesses in-context learning abilities with long-context examples from TREC (Li & Roth, 2002), TriviaQA (Joshi et al., 2017), and SAMSum (Gliwa et al., 2019).

- **Synthetic Tasks:** Includes PassageCount and PassageRetrieval-en to isolate specific long-range dependency capabilities.

- **Code Completion:** Evaluates programming context understanding using LCC (Chen et al., 2021) and RepoBench-P (Liu et al., 2024).

## D. Details about Baselines

We adopted the original hyperparameters for the baseline methods as reported in their respective papers, details in Table 8.

Table 8. Hyperparameter configurations for baseline methods, following their original reports.

| Method | Hyperparameters |
|---|---|
| SnapKV (Li et al., 2024) | Kernel size = 7, Window size = 32 |
| AdaKV (Feng et al., 2026b) | $\alpha_{safeguard} = 0.20$, Window size = 32 |
| PyramidKV (Cai et al., 2024) | $\beta = 20$, Window size = 8 |
| Expected Attention (Devoto et al., 2025) | $n_{future\_positions} = 512, n_{sink} = 4, \epsilon = 0.02$ |

## E. LU-KV Implementation Details

For offline calibration, we employed an AI-generated novel ($\approx 4{,}000$ words) paired with 30 generated questions. We utilized the $L_2$-norm of the projected Value vectors ($|vW_O|_2$) for scoring, applying intra-layer normalization to the results. We configured the attention sink size to 4. The recent token window was set to 1 for KeyDiff and maintained at 32 for SnapKV. Additionally, we imposed a maximum compression threshold of 99% for any attention head, ensuring a minimum retention rate of 1%.

## F. Additional Analyses

This section provides additional analyses on compression-ratio sensitivity, offline-profiling transferability, integration with other KV compression metrics, and comparison with HeadKV. Unless otherwise specified, all additional LongBench results in this section are reported on Llama-3.1-8B-Instruct, with Full Cache average performance of 49.29 under the same evaluation protocol as the main experiments.

### F.1. Compression-Ratio Sensitivity

Table 9 evaluates LU-KV under 50%, 80%, and 90% compression ratios on LongBench. LU-KV remains stable across these budgets and degrades more gracefully than PyramidKV and AdaKV under aggressive compression.

Table 9. Performance across compression ratios on LongBench. Full cache performance is 49.29.

| Method | $\sigma$=50% | $\sigma$=80% | $\sigma$=90% |
|---|---|---|---|
| PyramidKV (KeyDiff) | 45.71 | 40.57 | 36.56 |
| AdaKV (KeyDiff) | 48.55 | 45.09 | 38.21 |
| **LU-KV (KeyDiff)** | 48.47 | 47.16 | 44.29 |
| PyramidKV (SnapKV) | 46.65 | 42.36 | 38.02 |
| AdaKV (SnapKV) | 47.91 | 43.98 | 40.20 |
| **LU-KV (SnapKV)** | **49.38** | **47.70** | **44.69** |

### F.2. Transferability Analysis of offline Profiling Data

able 10 evaluates whether the offline profiling data transfers across different calibration corpora. The two profiling examples used in this analysis are listed below:

- **Chinese Novel (4k):** an AI-generated Chinese novel excerpt of approximately 4K tokens, paired with generated questions that query different information segments in the long context.
- **English SAT (150):** a much shorter English SAT Reading excerpt, e.g., a passage beginning with "The following text is from Edith Wharton's 1905 novel...", paired with SAT-style comprehension questions.

Both profiling examples are out-of-domain relative to LongBench. The comparison between the Chinese Novel profile and the English SAT profile indicates that LU-KV is not tightly coupled to a single profiling corpus, while still benefiting from calibration data that captures long-context retrieval behavior.

*Table 10.* Transferability analysis of offline profiling data on LongBench.

| Method | Compression Ratio($\sigma$) | Profiling Data | Avg. | Single-Doc QA | Multi-Doc QA | Summ. | Few-shot | Synthetic | Code |
|---|---|---|---|---|---|---|---|---|---|
| Full Cache | 0 | - | 49.29 | 43.43 | 46.33 | 28.91 | 69.53 | 53.99 | 58.01 |
| KeyDiff | 0.5 | - | 47.24 | 41.37 | 43.14 | 27.49 | 67.37 | **53.67** | 55.24 |
| LU-KV | 0.5 | Chinese Novel (4k) | 48.47 | 42.72 | 42.23 | 28.90 | **70.14** | 52.65 | 59.14 |
| LU-KV | 0.5 | English SAT (150) | **49.06** | **43.28** | **43.19** | **29.03** | 69.87 | 53.53 | **60.92** |
| KeyDiff | 0.8 | - | 41.37 | 32.16 | 34.86 | 24.48 | 60.77 | **54.35** | 48.17 |
| LU-KV | 0.8 | Chinese Novel (4k) | **47.16** | **41.85** | 40.25 | **27.89** | **67.33** | 53.04 | **58.26** |
| LU-KV | 0.8 | English SAT (150) | 46.67 | 40.66 | **40.80** | 27.26 | 66.77 | 53.10 | 57.06 |

## F.3. Comparison with HeadKV

We further compare LU-KV with HeadKV, an offline-calibration-based allocation method, under the original fixed-budget setting used by HeadKV. Table 11 reports the cross-task performance on LongBench. LU-KV improves the average score for both SnapKV and EA, indicating that metric-specific global allocation provides additional benefit under the same fixed-budget evaluation setting.

*Table 11.* Cross-task comparison with HeadKV on LongBench under the original HeadKV fixed-budget setting.

| Method | Avg. | Single-Doc QA | Multi-Doc QA | Summ. | Few-shot | Synthetic | Code |
|---|---|---|---|---|---|---|---|
| Full Cache | 49.29 | 43.43 | 46.33 | 28.91 | 69.53 | 53.99 | 58.01 |
| HeadKV (SnapKV) | 45.27 | 35.46 | 40.72 | 25.55 | 64.08 | 51.97 | 61.45 |
| LU-KV (SnapKV) | 46.37 | 37.99 | **42.57** | 26.45 | **65.02** | 51.04 | **61.87** |
| HeadKV (EA) | 45.34 | 37.79 | 39.74 | 26.88 | 63.53 | 51.09 | 59.70 |
| LU-KV (EA) | **47.12** | **41.32** | 41.75 | **27.34** | 64.77 | **53.23** | 60.99 |

## F.4. Integration with More KV Compression Metrics

We further evaluate whether LU-KV can be applied beyond the SnapKV and KeyDiff metrics used in the main experiments. Table 12 integrates LU-KV with KVZip and CAKE by keeping their underlying compression metrics while replacing the default budget allocation with LU-KV's global allocation. This separates the effect of the compression metric from the effect of budget allocation. Across both KVZip and CAKE, LU-KV improves the average score under the same compression ratio, with larger gains under more aggressive compression.

Table 13 summarizes the metric-level ablation across KVZip, CAKE, and EA. The purpose is to test whether LU-KV depends on a particular proxy metric, or whether it can improve different metrics once their offline utility profiles are available. The consistent gains across all three metrics suggest that LU-KV acts as a general allocation layer: it does not replace the underlying metric, but improves how the global KV budget is distributed for that metric.

*Table 12.* Integration with KVZip and CAKE on LongBench. The rows labeled LU-KV apply LU-KV's allocation to the corresponding metric.

| Method | Ratio | Avg. | Single-Doc QA | Multi-Doc QA | Summ. | Few-shot | Synthetic | Code |
|---|---|---|---|---|---|---|---|---|
| Full Cache | - | 49.29 | 43.43 | 46.33 | 28.91 | 69.53 | 53.99 | 58.01 |
| KVZip | 0.5 | 48.90 | 43.43 | 43.58 | 28.92 | 69.19 | 51.80 | 61.71 |
| LU-KV (KVZip) | 0.5 | **49.16** | 43.09 | 43.76 | 28.98 | 69.73 | 52.78 | 62.16 |
| KVZip | 0.8 | 47.41 | 43.77 | 41.41 | 28.48 | 66.55 | 49.21 | 59.80 |
| LU-KV (KVZip) | 0.8 | **48.81** | 43.31 | 42.57 | 28.67 | 68.58 | 52.50 | 63.25 |
| KVZip | 0.9 | 40.84 | 38.41 | 39.53 | 27.01 | 62.32 | 24.15 | 51.65 |
| LU-KV (KVZip) | 0.9 | **44.79** | 40.92 | 38.33 | 27.16 | 64.31 | 40.91 | 61.31 |
| CAKE | 0.5 | 47.24 | 42.33 | 41.17 | 27.12 | 66.71 | 53.14 | 58.83 |
| LU-KV (CAKE) | 0.5 | **49.73** | 43.38 | 46.34 | 28.66 | 69.87 | 54.48 | 60.97 |
| CAKE | 0.8 | 43.56 | 33.29 | 38.53 | 24.57 | 62.07 | 52.32 | 58.49 |
| LU-KV (CAKE) | 0.8 | **47.86** | 42.94 | 45.78 | 27.29 | 65.16 | 52.97 | 58.12 |
| CAKE | 0.9 | 39.93 | 25.39 | 34.26 | 22.61 | 60.09 | 47.55 | 58.38 |
| LU-KV (CAKE) | 0.9 | **45.36** | 36.88 | 43.98 | 24.49 | 62.90 | 52.96 | 57.58 |

*Table 13.* Metric ablation analysis on LongBench using LU-KV.

| Metric | Compression Ratio | Original Avg. | w/ LU-KV |
|---|---|---|---|
| Full Cache | - | 49.29 | - |
| KVZip | 0.5 | 48.90 | **49.16** |
| KVZip | 0.8 | 47.41 | **48.81** |
| KVZip | 0.9 | 40.84 | **44.79** |
| CAKE | 0.5 | 47.24 | **49.73** |
| CAKE | 0.8 | 43.56 | **47.86** |
| CAKE | 0.9 | 39.93 | **45.36** |
| EA | 0.5 | 48.35 | **49.37** |
| EA | 0.8 | 43.10 | **48.73** |
| EA | 0.9 | 36.58 | **46.28** |

