# OpenReview forum: "Predicting Future Utility: Global Combinatorial Optimization for Task-Agnostic KV Cache Eviction"
_ICML.cc/2026/Conference — ICML 2026 regular_

### Official Review · Reviewer_Zepk · 2026-02-26

**Soundness:** 3
**Presentation:** 3
**Significance:** 3
**Originality:** 3
**Overall Recommendation:** 4
**Confidence:** 4

**Summary:**

This paper pinpoint that previous budget allocation relies only on local information and overlooks the heterogeneity in predictive fidelity across attention heads. The authors propose that optimal budget allocations should be governed by the marginal utility of preserving long-term semantic information. They present LU-KV as a strategy to achieve better budget allocation. Experiments demonstrate clear advantages.

**Compliance With Llm Reviewing Policy:**

Affirmed.

**Final Justification:**

My previous point was that the authors should use the |vW| metric from Oracle Importance for evaluation, ensuring optimal consistency between analytical and experimental assessments, rather than relying on a future decoding window for eviction.

The author has, in fact, already done this. My prior comment stemmed from an oversight that Expected Attention already incorporates the |vW| metric from CriticalKV, which led to my misunderstanding. However, I suggest that the authors rename the “EA-style metric” in the experimental section to the CriticalKV metric, since CriticalKV introduced this metric and EA follows it. This would help prevent similar misunderstandings in the future.

Overall, both the paper and the response are satisfactory, and all my concerns have been addressed. I would maintain a positive assessment of this work.

**Key Questions For Authors:**

### Questions

I notice that the Oracle Importance definition uses its maximum potential contribution over a future decoding window, which somewhat overlaps with another work [2]. That work argues that this max-based approach captures worst-case risk during future decoding and thus improves performance. I want to know how the authors discuss the connection between adopting the same optimization tricks from different research perspectives.

[2] Taming the Fragility of KV Cache Eviction in LLM Inference

**Limitations:**

Yes

**Strengths And Weaknesses:**

### Strength

* The author identifies an interesting and reasonable point: current budget allocation strategies overlook the heterogeneity in predictive fidelity across attention heads.

* The LU-KV budget allocation strategy performs well under both the SnapKV-style  and the keydiff-style importance indicators.

* Beyond presenting improvements to their own method, the author uses layer eviction loss to analyze the limitations of prior approaches such as PyramidKV and AdaKV, which I consider important. This effectively visualizes the gains of the LU-KV strategy.

### Weakness

This paper appear novel and reasonable to me, but its weakness lies in its evaluation, which is not quite sufficient.

* The paper defines and observes Oracle Importance using the metric A|vW| from CriticalKV, yet the experiments rely only on simplified SnapKV-style and keydiff-style metrics. The authors should further evaluate the performance under the Oracle Importance indicator.

* Some offline-training based works, such as HeadKV [1], may not explicitly target long-term semantic information, but they could implicitly consider it during training. It would be helpful to include these methods in the comparison.

[1] Not All Heads Matter: A Head-Level KV Cache Compression Method with Integrated Retrieval and Reasoning

---

> ### Author Rebuttal · Authors · 2026-03-30
>
> We appreciate the reviewer's recognition of our work's novelty and technical soundness. Our point-by-point responses to the comments are provided below.
>
> **W1. ...the experiments rely only on simplified SnapKV-style and keydiff-style metrics.The authors should further evaluate the performance under the Oracle Importance indicator...**
>
> **R1.** We thank the reviewer for this insightful suggestion.  We would like to clarify the role of Oracle Importance in our framework: by definition, it quantifies the maximum potential contribution of a token within a future decoding window. Since this metric inherently depends on future query information, it serves as a **theoretical upper bound** to analyze the optimality gap, rather than a deployable online metric.
>
> In addition to SnapKV and KeyDiff presented in the main text, we have included results using the Expected Attention (EA) metric in Appendix. B.3, which show consistent improvements.
>
> Furthermore, we extended our evaluation to include the metrics utilized in KVZip and CAKE. The corresponding results, along with those for the EA metric, are presented in the table below.
>
> **Table 1: Metric ablation analysis on  LongBench**
> |Metric|Compression Ratio|Original Avg.|**w/ LU-KV**|**Improvement**|
> |:---|:---:|:---:|:---:|:---:|
> |**Full Cache**|-|49.29|-|-|
> |**KVZip**|0.5|48.90|**49.16**|+0.26|
> ||0.8|47.41|**48.81**|+1.40|
> ||0.9|40.84|**44.79**|+3.95|
> |**CAKE**|0.5|47.24|**49.73**|+2.49|
> ||0.8|43.56|**47.86**|+4.30|
> ||0.9|39.93|**45.36**|+5.43|
> |**EA**|0.5|48.35|**49.37**|+1.02|
> ||0.8|43.10|**48.73**|+5.63|
> ||0.9|36.58|**46.28**|+9.70|
>
> **W2. Some offline-training based works, such as HeadKV, may not explicitly target long-term semantic information, but they could implicitly consider it during training. It would be helpful to include these methods in the comparison.**
>
> **R2.** HeadKV[1] evaluates head importance through context retrieval and reasoning tasks to implicitly capture long-context information, then allocates budget proportionally based on head importance. Which is different with ours LU-KV. We summarize the three core technical distinctions between LU-KV and HeadKV in the following table.
>
> **Table 2: Comparison between LU-KV and HeadKV**
> |Dimension|LU-KV|HeadKV|
> |:---|:---|:---|
> |Calibration Data|Low sensitivity to task, data volume, and even language (see our response R2 to Reviewer 2)|Requires extensive synthetic datasets (~100 samples), covering context retrieval and reasoning tasks across diverse sequence lengths and needle injection depths|
> |Allocation Strategy|Models allocation as a marginal utility maximization problem, solved via convex-hull relaxation and a greedy solver|Allocate head-level budget based on retrieval- and reasoning-derived head importance, while requiring manual hyperparameter tuning for varying budgets|
> |Metric Adaptability|Computes customized, optimal allocations tailored to the specific underlying metric (e.g., SnapKV vs. KeyDiff)|Employs a static strategy where head budgets remain identical regardless of the underlying metric|
>
> To empirically demonstrate these differences, we compared HeadKV and LU-KV by strictly following the original evaluation settings of HeadKV, utilizing a fixed budget of 1024 tokens. The results are summarized in the table below.
>
> **Table 2: Cross-task performance comparison on LongBench**
> |Method|Avg. (Δ)|Single-Doc QA|Multi-Doc QA|Summarization|Few-shot|Synthetic|Code|
> |:---|:---|:---|:---|:---|:---|:---|:---|
> |Full Cache|49.29|43.43|46.33|28.91|69.53|53.99|58.01|
> |HeadKV (SnapKV)|45.27 (↓8.2%)|35.46|40.72|25.55|64.08|51.97|61.45|
> |LU-KV (SnapKV)|46.37 (↓5.9%)|37.99|**42.57**|26.45|**65.02**|51.04|**61.87**|
> |HeadKV (EA)|45.34 (↓8.0%)|37.79|39.74|26.88|63.53|51.09|59.70|
> |LU-KV (EA)|**47.12** (↓4.4%)|**41.32**|41.75|**27.34**|64.77|**53.23**|60.99|
>
> We will include this detailed comparison and discussion in the Related Work and Appendix sections of the revised manuscript.
>
> **Q1. ...the Oracle Importance definition uses its maximum potential contribution over a future decoding window, which somewhat overlaps with another work [2]...**
>
> **R3.** Fundamentally, DefensiveKV [2] is an aggregation strategy that optimizes for worst-case scenarios within individual attention heads. However, its 'worst-case' formulation is inherently focused on historical context, as it does not explicitly account for the evolving importance of tokens in future decoding steps. In contrast, LU-KV’s Oracle Importance aggregates across the temporal dimension of future decoding, identifying the maximum potential contribution across all future queries to establish a theoretical upper bound for long-horizon utility.
>
> In summary, DefensiveKV focuses on local aggregation stability within individual heads, whereas LU-KV globally optimizes budget allocation to maximize long-horizon utility across all heads.
>
> We will expand this discussion in the revised manuscript.

---

> > ### Author Rebuttal · Reviewer_Zepk · 2026-04-04
> >
> > Thank you for your response!
> >
> >
> >
> > However, I’m still not able to fully understand why the authors did not evaluate performance using the Oracle Importance indicator.  In particular, using the Oracle Importance metric \(A|vW|\) from CriticalKV seems to be a simple modification of SnapKV-style approach. This suggests that, in principle, a SnapKV-style method might only need to incorporate \(|vW|\) into the importance function.
> >
> >
> > Overall, the response is good, and I will maintain my positive score.

---

> > > ### Author Response · Authors · 2026-04-07
> > >
> > > We thank the reviewer for the helpful follow-up. We further clarify below why Oracle Importance is used as an analysis target rather than as a practical online metric.
> > >
> > > **Why Oracle Importance cannot be directly used by compression methods.** In the general KV cache compression setting, inference-time methods such as SnapKV and CriticalKV must make eviction decisions without access to future decoding states. However, the Oracle Importance in Eq. (4) is defined using attention weights over a **future decoding window**, where this window corresponds to the subsequent answer-generation process, i.e., the sequence of tokens produced until the full response is completed. Therefore, computing Oracle Importance requires access to attention patterns from future queries that have not yet occurred at the time of compression. For this reason, Oracle Importance cannot be directly incorporated into a deployable SnapKV-style or CriticalKV-style scoring rule.
> > >
> > > **The role of Oracle Importance in our framework.** Accordingly, we introduce Oracle Importance as an **analysis target**, rather than a deployable metric. Its purpose is to strictly measure the true contribution of each token during the decode phase, which allows us to quantify the optimality gap of heuristic metrics and to construct the offline profiling procedure in Sec. 4.3. Our main experiments therefore focus on practical heuristics such as SnapKV, KeyDiff, and EA under the query-agnostic setting. Moreover, as shown in Figure 2b, the optimal allocation trends are highly consistent across tasks, which enables us to estimate metric-specific budget profiles offline and apply them directly during online inference.
> > >
> > > **Oracle-based evaluation.** To directly address the reviewer's question, we additionally report results using the oracle-induced indicator at compression ratio 0.9. For clarity, this oracle evaluation is conducted **offline**: we first run full decoding to obtain the complete answer trajectory, then compute Eq. (4) using the resulting decode-phase attention weights to obtain oracle scores for ranking token importance, and finally evaluate retention under the same compression ratio. However, this is not a realistic inference-time setting, since it requires the future answer trajectory to be known in advance.
> > >
> > > **Table 1: Performance comparison on LongBench under compression ratio 0.9**
> > >
> > > | Metric | Avg. Score |
> > > | :--- | :---: |
> > > | Full Cache | 49.29 |
> > > | LU-KV (KeyDiff) | 44.29 |
> > > | LU-KV (SnapKV) | 44.69 |
> > > | LU-KV (EA) | 46.28 |
> > > | **Oracle** | **47.86** |
> > >
> > > We will revise the manuscript to make this distinction more explicit.

---

### Official Review · Reviewer_farV · 2026-03-09

**Soundness:** 3
**Presentation:** 3
**Significance:** 4
**Originality:** 3
**Overall Recommendation:** 4
**Confidence:** 4

**Summary:**

This paper proposes LU-KV, a global KV Cache eviction budget allocation strategy that precisely controls the KV cache size of each layer and head based on long-horizon marginal utility. LU-KV achieves a near-optimal allocation strategy by modeling a convex-hull relaxation problem with offline data profiling and a greedy solver, enabling a remarkable compression ratio with negligible performance loss.

**Compliance With Llm Reviewing Policy:**

Affirmed.

**Final Justification:**

The supplement has addressed my concerns, making the paper's evaluation stronger.

**Key Questions For Authors:**

1. Could the authors clarify where the 4K input context comes from in the evaluation setup? Also, how is the generation length determined (e.g., why 4K generation tokens), and have you considered other generation lengths? A brief justification or a sensitivity analysis would help.
2. In Figures 7–9, I’m confused by the curves labeled ‘mytest convex’ and ‘mytest mckp’. The dotted black line and the solid orange line appear to be swapped relative to the legend. Could the authors verify whether the legend/line styles are correct and fix any potential plotting/labeling issues?
3. Since the proposed method is based on budget allocation and the evaluation is question-agnostic, could the authors include a brief direct comparison against [1] and/or [2]? This would help position the contribution relative to the closest prior work.
4. Can small trade-off analysis on different compression ratios be conducted?

**Limitations:**

Yes.

**Strengths And Weaknesses:**

Strengths:

1. This paper proposed an allocation method based on novel theoretical analysis rather than empirical conclusions, which makes the idea convincing.
2. The paper is well-organized, and the writing is clear. The formula derivation in the analysis and methodology is detailed and understandable.
3. The extensive experiments on several benchmarks truly validate the effectiveness of the proposed algorithms.

Weaknesses:

1. The baseline methods are not very new. Several SOTA works have been proposed [1-2].
2. The data for offline profiling is constructed, but in the paper, the authors didn't mention the detailed information about the data.
3. Compression ratio trade-offs are not discussed. Only 50% (appendix) and 80% compression results are discussed.

[1] Qin, Ziran, et al. "CAKE: Cascading and Adaptive KV Cache Eviction with Layer Preferences." The Thirteenth International Conference on Learning Representations.

[2] Kim, Jang-Hyun, et al. "KVzip: Query-Agnostic KV Cache Compression with Context Reconstruction." The Thirty-ninth Annual Conference on Neural Information Processing Systems.

---

> ### Author Rebuttal · Authors · 2026-03-30
>
> We appreciate your thoughtful feedback. Please find our responses to your concerns below.
>
> **W1 & Q3. The baseline methods are not very new. Several SOTA...**
>
> **R1.** The exclusion of KVZip from our initial evaluation was due to its metric require twice prefill passes and its reliance on the AdaKV allocation strategy. As discussed in lines 379–383, this global allocation approach prevents the reduction of peak VRAM. Both the two factors significantly increase inference latency.
>
> To provide a more comprehensive evaluation, we integrate LU-KV with KVZip and CAKE in the following tables.
>
> **Table 1: Integration with KVZip on LongBench**
> |Method|Ratio|Avg. (Δ)|Single-Doc QA|Multi-Doc QA|Summarization|Few-shot|Synthetic|Code|
> |:---|:---|:---|:---|:---|:---|:---|:---|:---|
> |Full Cache|-|49.29|43.43|46.33|28.91|69.53|53.99|58.01|
> |KVZip|0.5|48.90 (↓0.79%)|43.43|43.58|28.92|69.19|51.80|61.71|
> |LU-KV (KVZip)|0.5|**49.16** (↓0.26%)|43.09|43.76|28.98|69.73|52.78|62.16|
> |KVZip|0.8|47.41 (↓3.80%)|43.77|41.41|28.48|66.55|49.21|59.80|
> |LU-KV (KVZip)|0.8|**48.81** (↓0.98%)|43.31|42.57|28.67|68.58|52.50|63.25|
> |KVZip|0.9|40.84 (↓17.15%)|38.41|39.53|27.01|62.32|24.15|51.65|
> |LU-KV (KVZip)|0.9|**44.79** (↓9.13%)|40.92|38.33|27.16|64.31|40.91|61.31|
>
> **Table 2: Integration with CAKE on LongBench.**
> CAKE innovates in both metrics and allocation strategy. By applying LU-KV’s allocation to CAKE’s metrics, LU-KV consistently outperform CAKE’s default strategy
> |Method|Ratio|Avg. (Δ)|Single-Doc QA|Multi-Doc QA|Summarization|Few-shot|Synthetic|Code|
> |:---|:---|:---|:---|:---|:---|:---|:---|:---|
> |Full Cache|-|49.29|43.43|46.33|28.91|69.53|53.99|58.01|
> |CAKE|0.5|47.24 (↓4.15%)|42.33|41.17|27.12|66.71|53.14|58.83|
> |LU-KV (CAKE)|0.5|**49.73** (↑0.89%)|43.38|46.34|28.66|69.87|54.48|60.97|
> |CAKE|0.8|43.56 (↓11.62%)|33.29|38.53|24.57|62.07|52.32|58.49|
> |LU-KV (CAKE)|0.8|**47.86** (↓2.91%)|42.94|45.78|27.29|65.16|52.97|58.12|
> |CAKE|0.9|39.93 (↓18.99%)|25.39|34.26|22.61|60.09|47.55|58.38|
> |LU-KV (CAKE)|0.9|**45.36** (↓7.96%)|36.88|43.98|24.49|62.90|52.96|57.58|
>
> **W2 & Q1. ...in the paper, the authors didn't mention the detailed information about the offline profiling data...why 4K generation tokens...**
>
> **R2.** Our default profiling utilizes an AI-generated Chinese novel (~4K tokens), yet it demonstrates significant cross-lingual and cross-domain transferability when evaluated on English benchmarks (Document QA, Summarization, Code, Multi-hop reasoning, etc).
>
> Chinese novel excerpt：
> > {"context":"小说策划案：《第十三次谢幕》\n\n第一章：暴风雪中的请柬\n\n暴风雪将盘山公路吞没。...","questions":[ "...人物塑造中起到了什么作用？"]}
>
> Our choice of a 4K context was primarily motivated by the need to maximize A100 (80GB) GPU utilization. To assess our method's versatility, we further conducted a transferability test using a distinct, short-form English SAT Reading sample (~150 tokens).
>
> Englis SAT Reading excerpt:
> > {"context": "The following text is from Edith Wharton’s 1905 novel...", "questions": ["Which choice best describes...A...]}
>
> As shown in the table below, LU-KV remains highly effective regardless of the profiling corpus's language or length, significantly outperforming the baseline in broad tasks.
>
> **Table 2: Transferability Analysis of Offline Profiling Data.**
> |Method|Ratio|Profiling Data|Avg. (Δ)|Single-Doc QA|Multi-Doc QA|Summarization|Few-shot|Synthetic|Code|
> |:---|:---|:---|:---|:---|:---|:---|:---|:---|:---|
> |Full Cache|0|-|49.29|43.43|46.33|28.91|69.53|53.99|58.01|
> |KeyDiff|0.5|-|47.24 (↓4.16%)|41.37|43.14|27.49|67.37|**53.67**|55.24|
> |LU-KV|0.5|Chinese Novel (4k)|48.47 (↓1.67%)|42.72|42.23|28.90|**70.14**|52.65|59.14|
> |LU-KV|0.5|English SAT (150)|**49.06** (↓0.47%)|**43.28**|**43.19**|**29.03**|69.87|53.53|**60.92**|
> |KeyDiff|0.8|-|41.37 (↓16.0%)|32.16|34.86|24.48|60.77|**54.35**|48.17|
> |LU-KV|0.8|Chinese Novel (4k)|**47.16** (↓4.32%)|**41.85**|40.25|**27.89**|**67.33**|53.04|**58.26**|
> |LU-KV|0.8|English SAT (150)|46.67 (↓5.31%)|40.66|**40.80**|27.26|66.77|53.10|57.06|
>
> **W3 & Q4. Compression ratio trade-offs are not discussed. Only 50% (appendix) and 80% compression...**
>
> **R3.** We report the results for 50%, 80%, and 90% compression ratios in the following table. At a 50% ratio, LU-KV (SnapKV) achieves near-lossless (and in some cases, improved) performance. For more aggressive pruning, the 80% ratio serves as an optimal balance between efficiency and accuracy for our method.
>
> **Table 1: Performance across compression ratios on LongBench.**
> |Method|50%|80%|90%|
> |:---|:---|:---|:---|
> |Ada（KeyDiff）|48.55|45.09|38.21|
> |LU-KV（KeyDiff）|48.47|47.16|44.29|
> |Ada（SnapKV）|47.91|43.98|40.20|
> |LU-KV（SnapKV）|**49.38**|**47.70**|**44.69**|
>
> **Q2. ...confused by the curves labeled ‘mytest convex’ and ‘mytest mckp’...**
>
> **R5.** The black solid line (mytest mckp) is currently obscured by the orange dashed line (mytest convex), giving it the appearance of a dashed line. To improve clarity, we will update the legend and represent 'mytest mckp' with a black dashed line instead.

---

> > ### Author Rebuttal · Reviewer_farV · 2026-04-02
> >
> > The supplement has addressed my concerns, making the paper's evaluation stronger. Score increased.

---

> > > ### Author Response · Authors · 2026-04-07
> > >
> > > Thank you for your review and for increasing your score. We are pleased that the additional details in the supplement resolved your concerns. Your helpful comments provided a great perspective to make our evaluation more comprehensive. We sincerely appreciate your time and support.

---

### Official Review · Reviewer_tP5z · 2026-03-12

**Soundness:** 3
**Presentation:** 2
**Significance:** 3
**Originality:** 3
**Overall Recommendation:** 4
**Confidence:** 4

**Summary:**

This paper studies KV-cache eviction for decoder-only LLMs. It argues that common eviction heuristics based on instantaneous proxy scores (e.g. attention) are misaligned with tokens’ long-horizon utility during future decoding. The authors introduce a long-horizon utility framing, define an “optimality gap” between proxy-driven selections and an oracle notion of importance, and cast head-level cache budgeting as a global constrained optimization problem. They propose an efficient allocation procedure and an offline profiling protocol that enables low-overhead inference-time execution. Experiments on LongBench and RULER indicate strong performance at aggressive compression (eg 80%).

**Compliance With Llm Reviewing Policy:**

Affirmed.

**Final Justification:**

Rebuttal answered all my concerns, and now the paper has stronger results which provide significant novelty.

**Key Questions For Authors:**

1. **Sigma / budget definition (related to my clarity concern):** Is sigma the fraction *evicted* or *retained*? Please give the explicit mapping from sigma to `B_total` and `b_{l,h}`, and point to where sigma enters the optimization/solver.

2. **Compression sweep (related to my robustness concern):** Do your gains persist across multiple compression ratios on LongBench and RULER? Please report quality-vs-budget curves for your main baselines.

3. **Offline profiling transfer (related to my profiling/shift concern):** What exact profiling dataset/prompts and decoding settings are used to build the lookup table, and how well does a table profiled on one domain transfer to other domains/context lengths?

**Limitations:**

No.

The paper includes an impact statement, but the limitations discussion could be more explicit. In particular, it would help to briefly discuss the dependence on offline profiling, how well the profiled policy transfers across tasks or context regimes, and how sensitive the method is to the chosen compression ratio. A short note on practical deployment tradeoffs would also strengthen this section.

**Strengths And Weaknesses:**

## Strengths
- **Significance (problem relevance):** KV-cache eviction is a practically important bottleneck for long-context LLM inference; improving retention decisions under a hard KV budget can have clear deployment value.
- **Good motivation and framing:** The mismatch between short-horizon proxy metrics and long-horizon utility is real, and the cross-head budgeting view is sensible.
- **Originality (framing + solver):** Casting head-wise budgeting as a global constrained optimization (and proposing an efficient allocation procedure + offline profiling/lookup deployment) is a coherent combination that goes beyond typical heuristic eviction rules.
- **Empirical results look promising** in the shown regime (notably at high compression).

## Weaknesses
- **Soundness (evidence gaps):** The paper emphasizes results at a single aggressive point (80% compression). Without a sweep over multiple compression ratios, it is hard to assess robustness of the tradeoff and whether the chosen ratio is a sweet spot.
- **Soundness (profiling validity):** Since the approach relies on offline profiling, the paper should more clearly evaluate transfer under domain shift (different tasks, distributions, context lengths, decoding params). Otherwise the method risks being brittle or requiring frequent re-profiling.

- **Presentation issues hurt readability.** There is too much bolding, which makes sections harder to read as coherent paragraphs.

- **Missing or delayed definitions.** Key symbols/parameters appear before they are defined, which forces backtracking and makes Figure 1/2 hard to interpret.
  - `T`: used early (eg indices in `{1,...,T}`) but not defined locally when first introduced.
  - `sigma` (target compression ratio): appears as the key knob, but its meaning (evicted vs retained) and mapping to budgets is not stated at first use.
  - `b_{l,h}` vs `B_total`: the relationship between per-head budgets and the global budget is introduced later but earlier text implicitly relies on it.
  - “Local compression ratio”: referenced (eg around Figure 2a) without an operational definition (per-head? per-layer? per-step?).

- **Notation inconsistency / ambiguity.** Some indices and set expressions are hard to follow, and a few equations look inconsistent on first read.
  - Index/set membership mismatch: cases where an index (eg `j`) appears to belong to a set in one equation but not in the corresponding summation/definition in another.
  - Ambiguity around `e` / `j`: unclear scoping and meaning across equations.
  - `M^pi_{l,h}(b_{l,h})`: needs a plain-English explanation in KV-cache terms (what is kept vs evicted), otherwise later hit/miss/fp partitions read as symbol pushing.
  - Table semantics: “Full-KV” appears alongside metric blocks (`pi_1`/`pi_2`) but is not explicitly defined (no-eviction upper bound vs something else).

---

> ### Author Rebuttal · Authors · 2026-03-30
>
> Thank you for your valuable feedback. Our responses to your questions are provided below.
>
> **W1 & Q2. The paper emphasizes results at a single aggressive point (80% compression).**
>
> **R1.** We would like to clarify that our original submission included results for **both 80% and 50% compression ratios**; the latter was placed in the Appendix B.3 due to the page limitation.
>
> We futher provide results under more compression rate in the following table.
>
>
> **Table 1: Performance across compression ratios on LongBench. Full Cache is 49.29.**
> |Method|50%|80%|90%|
> |:---|:---|:---|:---|
> |Pyramid（KeyDiff）|45.71|40.57|36.56|
> |Ada（KeyDiff）|48.55|45.09|38.21|
> |**LU-KV**（KeyDiff）|48.47|47.16|44.29|
> |Pyramid（SnapKV）|46.65|42.36|38.02|
> |Ada（SnapKV）|47.91|43.98|40.20|
> |**LU-KV**（SnapKV）|**49.38**|**47.70**|**44.69**|
>
> At a 50% ratio, LU-KV (SnapKV) achieves near-lossless (and in some cases, improved) performance. For more aggressive pruning, the 80% ratio serves as an optimal balance between efficiency and accuracy for our method.
>
> **W2 & Q3. Generalization and Transferability of Offline Profiling**
>
> **R2.** Our default profiling utilizes an AI-generated Chinese novel (4K tokens), yet it demonstrates significant cross-lingual and cross-domain transferability when evaluated on English benchmarks (Document QA, Summarization, Code, Multi-hop reasoning, etc).
>
> Offline profiling data - Chinese novel excerpt (~ 4K tokens)：
> > {"context":"小说策划案：《第十三次谢幕》\n\n第一章：暴风雪中的请柬\n\n暴风雪将盘山公路吞没。...","questions":[ "...人物塑造中起到了什么作用？"]}
>
> To further validate this, we conducted a transferability test using a drastically different and smaller corpus: an English SAT Reading sample (~150 tokens).
>
> Offline profiling data - English SAT Reading excerpt (~150 tokens):
> > {"context": "The following text is from Edith Wharton’s 1905 novel...", "questions": ["Which choice best describes...A...]}
>
> In both cases, the profiling data used is out-of-domain relative to our evaluation benchmark.
> As shown in the table below, LU-KV remains highly effective regardless of the profiling corpus's language or length, significantly outperforming the baseline in broad tasks.
>
> **Table 2: Transferability Analysis of Offline Profiling Data.**
> |Method|Ratio|Profiling Data|Avg. (Δ)|Single-Doc QA|Multi-Doc QA|Summarization|Few-shot|Synthetic|Code|
> |:---|:---|:---|:---|:---|:---|:---|:---|:---|:---|
> |Full Cache|0|-|49.29|43.43|46.33|28.91|69.53|53.99|58.01|
> |KeyDiff|0.5|-|47.24 (↓4.16%)|41.37|43.14|27.49|67.37|**53.67**|55.24|
> |LU-KV|0.5|Chinese Novel (4k)|48.47 (↓1.67%)|42.72|42.23|28.90|**70.14**|52.65|59.14|
> |LU-KV|0.5|English SAT (150)|**49.06** (↓0.47%)|**43.28**|**43.19**|**29.03**|69.87|53.53|**60.92**|
> |KeyDiff|0.8|-|41.37 (↓16.0%)|32.16|34.86|24.48|60.77|**54.35**|48.17|
> |LU-KV|0.8|Chinese Novel (4k)|**47.16** (↓4.32%)|**41.85**|40.25|**27.89**|**67.33**|53.04|**58.26**|
> |LU-KV|0.8|English SAT (150)|46.67 (↓5.31%)|40.66|**40.80**|27.26|66.77|53.10|57.06|
>
> **W3. Presentation issues - too much bolding**
>
> **R3.** We thank the reviewer for the feedback regarding formatting. We will reduce the use of bold text in the revised manuscript to improve readability.
>
> **W4, W5 & Q1. Missing or delayed definitions, and Notation inconsistency / ambiguity.**
>
> **R4.** We have provided a **summary of all notations in a dedicated table (Lines 125–134)**. Below are some definitions:
> * $T$: The total number of input tokens during the prefill phase.
> * $B_{total}$: Global memory budget (total number of KV pairs to retain).
> * $b_{l,h}$: Specific memory budget allocated to the attention head at layer $l$ and index $h$.
> * $M^\pi_{l,h}(k)$: The pruned subset containing the top $k$ elements according to policy $\pi$ for head $(l, h)$.
> * $\sigma$: Compression ratio (the percentage of tokens to be evicted).
>
> We will move this notation table to a more prominent position so that it precedes any notation. And we will refine our notation in the revised manuscript to resolve any ambiguity:
> * Local compression ratio: head-level compression ratio.
> * Index $j$: We will standardize the use of indices for token positions to avoid confusion.
> * Full-KV: This term refers to the baseline where no KV cache compression is applied (retaining all tokens). We will ensure this is explicitly defined in the experiment setup.
>
> **Q1. Is sigma the fraction evicted or retained?...give the explicit mapping from sigma to $B_{total}$ and $b_{l,h}$...**
>
> **R5.** Please refer to R4 for the formal definition of $\sigma$. To clarify the mapping: the global memory budget is defined as $B_{total} = (1-\sigma)T$. The specific allocation for each head, $b_{l,h}$, is then derived by solving the optimization objective formulated in Eq. (10).
>
>
> **Limitation: ...the limitations discussion could be more explicit...**
>
> **R6.** We will incorporate the sensitivity analysis regarding our profiling data and the trade-offs associated with different compression ratios, as discussed above, into the final manuscript's impact statement.

---

> > ### Author Rebuttal · Reviewer_tP5z · 2026-03-31
> >
> > Table 1 and 2 provide significant ablations, making the paper's novelty stronger.
> > I have raised my score accordingly.

---

> > > ### Author Response · Authors · 2026-04-07
> > >
> > > Thank you for your time, the positive feedback, and for raising your score. We are glad that the ablation studies in Tables 1 and 2 effectively addressed your concerns and helped better demonstrate the novelty of our work. We appreciate your constructive suggestions, which guided us to further polish the manuscript.

---

### Decision · Program_Chairs · 2026-04-30

**Decision:**

Accept (regular)

**Comment:**

This paper addresses a critical bottleneck in long-context Large Language Model (LLM) inference: the quadratic complexity of attention and the resulting memory overhead of the Key-Value (KV) cache. The authors propose LU-KV, a novel framework that shifts away from traditional instantaneous heuristic metrics for cache eviction. Instead, it treats head-level budget allocation as a global constrained optimization problem governed by the marginal utility of preserving long-term semantic information.

All reviewers support accepting this papers, highlighting the merit of the paper:

1) Reviewers praised the shift from empirical heuristics to a solid theoretical foundation for budget allocation. The cross-head budgeting view was deemed highly sensible and original.

2) While initial concerns were raised regarding evaluation at a single compression ratio (80%), the authors successfully provided a comprehensive sweep across 50%, 80%, and 90% ratios during the rebuttal. LU-KV (SnapKV) achieved near-lossless performance at 50% compression.

3) The authors addressed concerns regarding the "offline" nature of the profiling by demonstrating strong cross-lingual and cross-domain transferability. Profiling on a Chinese novel effectively transferred to English benchmarks including Document QA, Summarization, and Code tasks.

4) The rebuttal effectively integrated LU-KV with existing state-of-the-art methods like KVZip, CAKE, and HeadKV, consistently showing that LU-KV’s allocation strategy outperforms default heuristics.

The authors have addressed technical concerns regarding notation, baseline comparisons, and the practicalities of "Oracle Importance" versus online metrics. The paper presents a technically sound and well-motivated advancement in LLM efficiency that is likely to impact future research in long-context modeling. The AC doesn't find significant reasons to overturn reviewers recommendations and thus suggest accepting the paper.